# Controllable ion transport by surface-charged graphene oxide membrane

Mengchen Zhang[1], Kecheng Guan[1], Yufan Ji[1], Gongping Liu[1], Wanqin Jin[1] & Nanping Xu[1]

Ion transport is crucial for biological systems and membrane-based technology. Atomic-thick two-dimensional materials, especially graphene oxide (GO), have emerged as ideal building blocks for developing synthetic membranes for ion transport. However, the exclusion of small ions in a pressured filtration process remains a challenge for GO membranes. Here we report manipulation of membrane surface charge to control ion transport through GO membranes. The highly charged GO membrane surface repels high-valent co-ions owing to its high interaction energy barrier while concomitantly restraining permeation of electrostatically attracted low-valent counter-ions based on balancing overall solution charge. The deliberately regulated surface-charged GO membranes demonstrate remarkable enhancement of ion rejection with intrinsically high water permeance that exceeds the performance limits of state-of-the-art nanofiltration membranes. This facile and scalable surface charge control approach opens opportunities in selective ion transport for the fields of water transport, biomimetic ion channels and biosensors, ion batteries and energy conversions.

---

[1] State Key Laboratory of Materials-Oriented Chemical Engineering, Jiangsu National Synergetic Innovation Center for Advanced Materials, College of Chemical Engineering, Nanjing Tech University, 5 Xinmofan Road, 210009 Nanjing, P.R. China. Correspondence and requests for materials should be addressed to G.L. (email: gpliu@njtech.edu.cn) or to W.J. (email: wqjin@njtech.edu.cn)

A common natural phenomenon "like charges repel while unlike charges attract" is the rule of ion transport that is essential to our daily life[1]. In biological membranes, selectivity filter of an ion channel is enriched by charged residues, which enable biological systems to achieve ultrahigh efficiency while displaying selectivity for transmembrane ion transport[2–4]. These fascinating properties of biological membranes have motivated researchers to exploit synthetic membranes with ionic transport channels that have received particular attention in the selective removal of salts from water to produce industrial soft water and potable water[5,6]. Graphene oxide (GO) membrane is expected to share structural features with biological membrane owing to its water-transport pathways through assembled GO laminates, which has generated immense interest from the scientific community to study its transport properties and

mechanisms[7–11]. Computer simulations[12] and self-diffusion measurements[13,14] have demonstrated that specific ions could selectively permeate through GO membrane mainly based on size sieving effect and interaction between ions and GO membrane, while the high-throughput manufacture and industrial implementation of these GO membranes need to be further studied to validate their practicality and scalability. However, in pressure-driven filtration processes, GO membranes generally failed to selectively transport ions, mainly because their interlayer spacing was too large to sieve ions especially when GO membranes were swollen in water[15,16]. Numerous attempts to modulate the interlayer spacing of GO membranes have been undertaken by partial reduction[17], cross-linking[18], and building multilayer architectures[19,20]. Precise tuning GO interlayer spacing within subnanometer range is challenging[21] and remains difficult

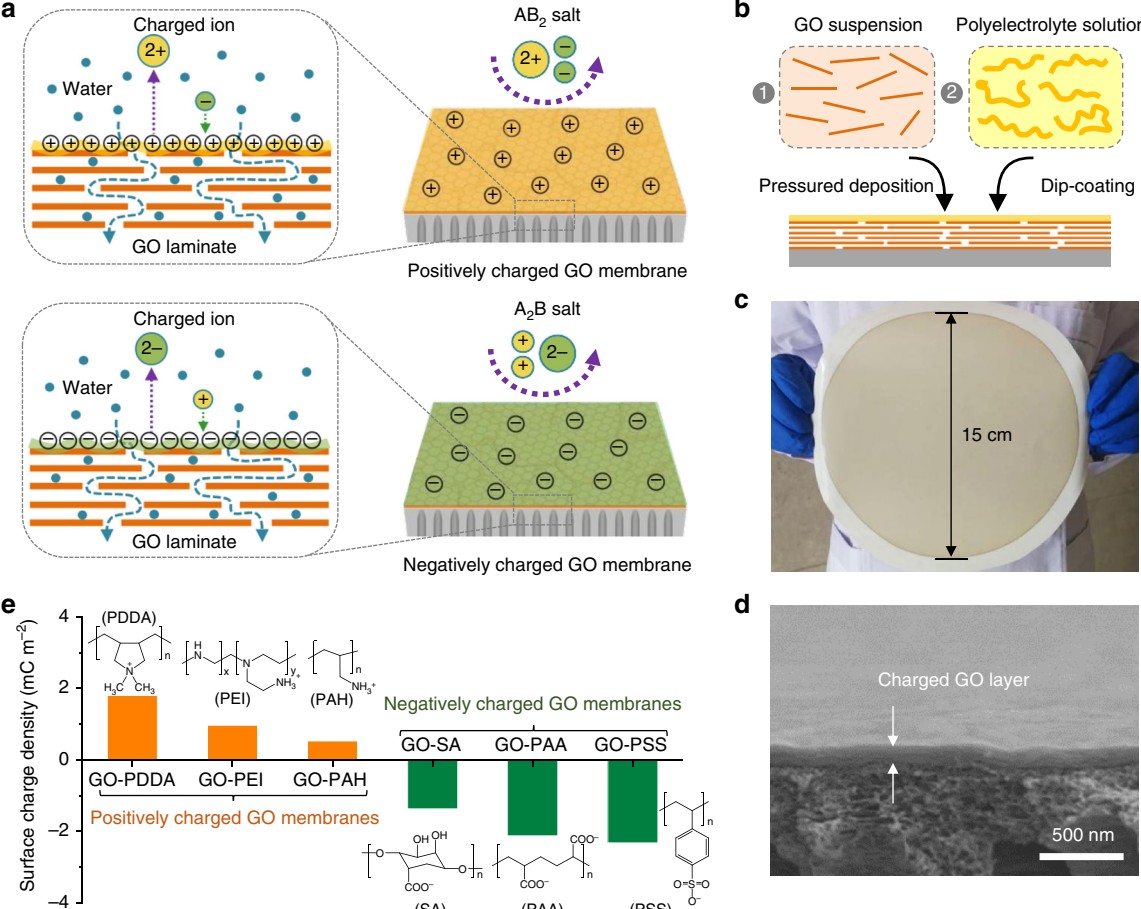

**Fig. 1** Design of surface-charged graphene oxide (GO) membrane. **a** Schematic of the design of surface-charged GO membranes by coating polyelectrolytes on the surface of GO laminates to realize controllable ion transport. Coating polycations such as polydiallyl dimethyl ammonium (PDDA), polyethylene imine (PEI), and polyallylamine hydrochloride (PAH) led the GO membrane to exclude $AB_2$-type salts based on the positively charged membrane surface, which exhibits a dominant electrostatic repulsion against divalent cations $A^{2+}$, which is favored over electrostatic attraction with monovalent anions $B^-$; coating polyanions such as polystyrene sulfonate (PSS), polyacrylic acid (PAA), and sodium alginate (SA) led the GO membrane to exclude $A_2B$-type salts based on the negatively charged membrane surface, which exhibits a dominant electrostatic repulsion against divalent anions $B^{2-}$, which is favored over electrostatic attraction with monovalent cations $A^+$. **b** Schematic of the preparation of surface-charged GO membranes. GO laminates were first prepared by filtrating GO aqueous suspension on a porous polyacrylonitrile (PAN) substrate via pressured-assisted filtration–deposition method, followed by dip-coating a dilute polyelectrolyte solution on surface of pre-stacked GO laminates to form the surface-charged GO membranes. **c** Photograph of large-area surface-charged GO membrane (GO deposition amount of 5 mg with 0.1 wt% PDDA polyelectrolyte surface coating) with a diameter of 15 cm (effective area: ~180 cm$^2$). **d** Scanning electron microscopic cross-sectional views of surface-charged GO membranes on top of a porous PAN substrate (GO deposition amount of 0.5 mg with 0.1 wt% PEI polyelectrolyte surface coating; membrane diameter of 4.7 cm with effective area of ~17.35 cm$^2$). **e** Surface charge densities of surface-charged GO membranes calculated from the measured membrane zeta potentials based on Gouy–Chapman theory. Insets are molecular structures of the surface polyelectrolytes with ionized functional groups

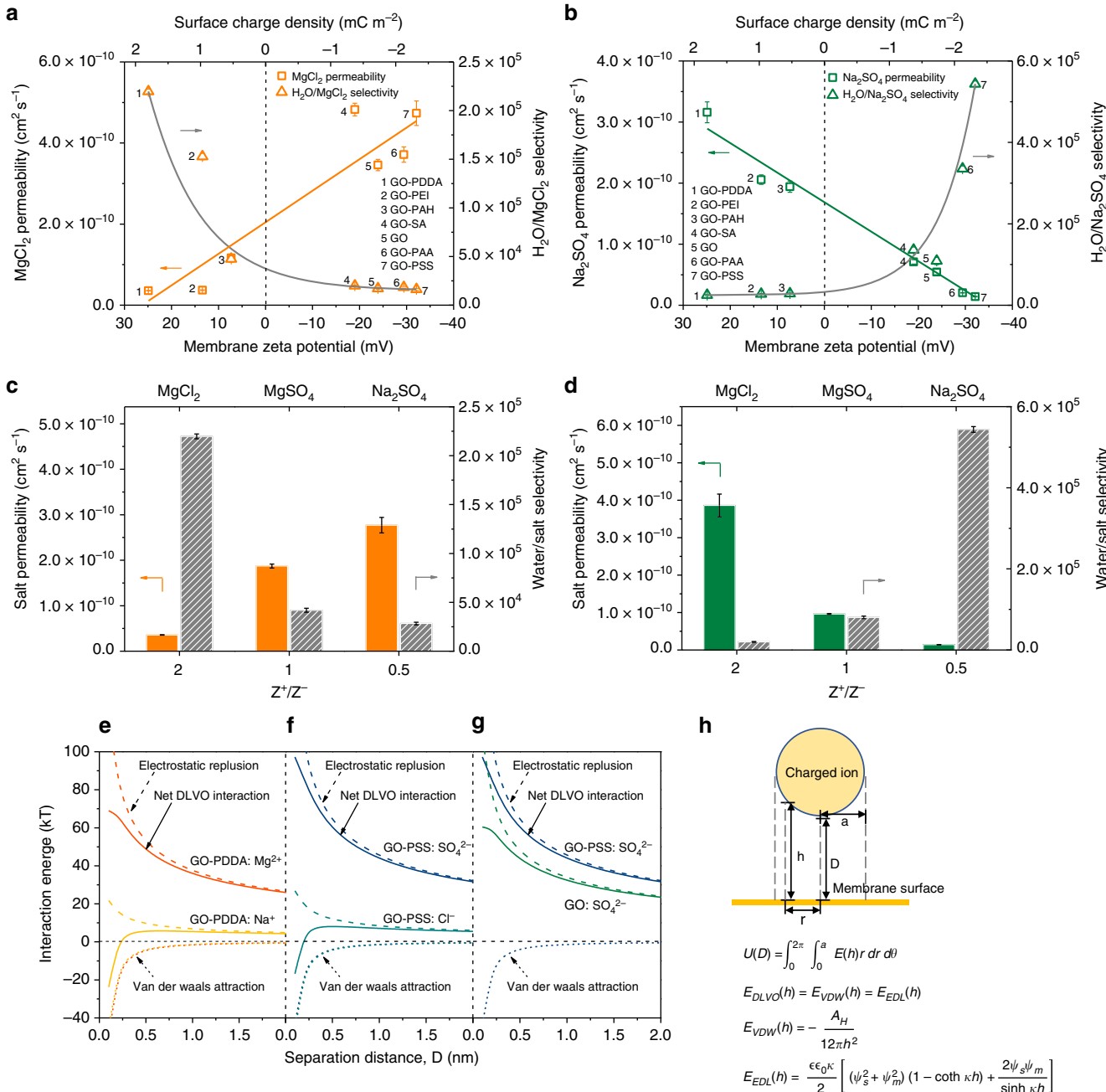

**Fig. 2** Ion transport mechanism of surface-charged graphene oxide (GO) membranes. **a** $MgCl_2$ permeability and $H_2O/MgCl_2$ selectivity and **b** $Na_2SO_4$ permeability and $H_2O/Na_2SO_4$ selectivity of surface-charged GO membranes with various membrane zeta potentials obtained by streaming potential measurements and surface charge densities calculated from Gouy–Chapman equation. Orange squares are $MgCl_2$ permeability and orange circles are $H_2O/MgCl_2$ selectivity; Green squares are $Na_2SO_4$ permeability and green circles are $H_2O/Na_2SO_4$ selectivity; Solid lines are best fits for the data. **c**, **d** Salt permeability and water/salt selectivity of **c** positively charged and **d** negatively charged GO membranes for $MgCl_2$, $MgSO_4$, and $Na_2SO_4$ salts with varied $Z^+/Z^-$ (ratio of the valence of cation and anion) values. **e**–**g** Surface element integration model predictions of Derjaguin–Landau–Verwey–Overbeek (DLVO) interaction energies between a charged ion and the charged membrane surface by adding Van der Waals attraction and electrostatic repulsion. **e** Orange: net DLVO interaction energy (solid line), electrostatic repulsion (dashed line), and Van der Waals attraction (dotted line) between $Mg^{2+}$ and the GO-PDDA membrane; Yellow: net DLVO interaction energy (solid line), electrostatic repulsion (dashed line), and Van der Waals attraction (dotted line) between $Na^+$ and the GO-PDDA membrane; **f** Navy: net DLVO interaction energy (solid line), electrostatic repulsion (dashed line), and Van der Waals attraction (dotted line) between $SO_4^{2-}$ and the GO-PSS membrane; Blue: net DLVO interaction energy (solid line), electrostatic repulsion (dashed line), and Van der Waals attraction (dotted line) between $Cl^-$ and the GO-PSS membrane; **g** Navy: net DLVO interaction energy (solid line), electrostatic repulsion (dashed line), and Van der Waals attraction (dotted line) between $SO_4^{2-}$ and the GO-PSS membrane; Green: net DLVO interaction energy (solid line), electrostatic repulsion (dashed line), and Van der Waals attraction (dotted line) between $SO_4^{2-}$ and the GO membrane. **h** Calculation formulas and schematic of the surface element integration (SEI) model in the calculation for DLVO interactions. Error bars represent standard deviations for three measurements

to achieve high salt rejection in a pressured filtration process, which furthermore sacrifices the intrinsically fast water permeation through the interlayer channels.

Herein, inspired by the charge interaction principle and the function of biological ion channels, we demonstrate a strategy of creating surface charges on GO membrane to realize controllable ion transport without impeding water filtration though the GO membrane. Tunable charges attached on the surface of pre-stacked GO laminates exhibited dominant electrostatic repulsion against doubly charged co-ions (with charge like the membrane surface charge) while suppressing weak electrostatic attraction toward singly charged counter-ions (with charge unlike the membrane surface charge). By simply manipulating the charge interactions between the membrane surface and the ions in water, transport of ions from typical AB$_2$- or A$_2$B-type salts can be prevented while the water remains free to permeate through the membrane (Fig. 1a).

## Results

**Manipulation of GO membrane surface charge.** First, we prepared stabilized GO laminates using low O/C ratio (0.186) GO materials on top of a porous surface hydrolyzed polyacrylonitrile (h-PAN) substrate with strong interfacial adhesion[22], followed by the attachment of tunable surface charges via dip-coating an array of selected polyelectrolytes (polycations: polydiallyl dimethyl ammonium (PDDA), polyethylene imine (PEI), polyallylamine hydrochloride (PAH); polyanions: polystyrene sulfonate (PSS), polyacrylic acid (PAA), sodium alginate (SA)) on the surface of pre-stacked GO laminates to create surface-charged GO membranes (Fig. 1b and Supplementary Fig. 1). Representative photos of a typical membrane and its morphology are displayed in Fig. 1c, d, showing a uniform, large-area (~15 cm in diameter) membrane with a thin, defect-free charged GO layer of ~100 nm thickness (Supplementary Figs 2 and 3). The as-prepared surface-charged GO membrane preserved the laminar structure of the GO laminate (Supplementary Fig. 4) with an ultrathin polyelectrolyte layer (Supplementary Figs 5, 6 and Table 1) firmly integrated on the surface via hydrogen binding and/or electrostatic attraction (Supplementary Figs 7, 8 and 10a). X-ray photoelectron spectroscopy (XPS; Supplementary Fig. 8) and infrared (IR; Supplementary Fig. 9) spectra of surface-charged GO membranes showed newly introduced functional groups on the GO membrane surface derived from the top polyelectrolytes. Protonation of amine groups or deprotonation of sulfonic/carboxyl/hydroxyl groups in water accounted for the charge properties of the membrane surface, which could be finely tuned by the intensity and amount of these ionizable functional groups. As quantified by membrane surface charge density[23], PDDA with stronger protonation than PEI and PAH produced most positively charged GO-PDDA membrane with surface charge density of $+1.8$ mC m$^{-2}$; likewise, the most negatively charged GO-PSS membrane with surface charge density of $-2.32$ mC m$^{-2}$ resulted from the attached PSS with the highest deprotonation than PAA and SA (Fig. 1e). An identical order of membrane surface charge density with respect to the zeta potential of the attached polyelectrolytes (Supplementary Fig. 10) suggests that the charge properties of polyelectrolytes can be easily translated onto GO membrane surface via a simple coating method.

**Ion transport behavior through surface-charged GO membrane.** To explore the roles of membrane surface charges, we investigated ion transport behavior through the surface-charged GO membranes by evaluating salt permeability and water/salt selectivity[23] in filtration measurements using the model of saline containing MgCl$_2$ or Na$_2$SO$_4$ (Fig. 2a, b). Tuning the membrane surface charge from highly positive to highly negative, MgCl$_2$ permeability showed a continuous increase while Na$_2$SO$_4$ permeability showed an

approximately linear reduction. Consequently, extraordinarily high H$_2$O/MgCl$_2$ selectivity of $2.2 \times 10^5$ was achieved in the highly positively charged GO membrane (GO-PDDA), but it underwent an exponential decay of more than an order of magnitude as the membrane became negatively charged (GO-PSS). Conversely, the relatively low H$_2$O/Na$_2$SO$_4$ selectivity of the highly positively charged GO membrane (GO-PDDA) was rapidly promoted by over 20-fold, reaching $5.4 \times 10^5$ as the membrane surface charge was tuned to a highly negative value (GO-PSS). The exactly reverse trends led to the hypothesis that a positively charged GO membrane would tend to prevent transport of an AB$_2$ salt containing the divalent cation (A$^{2+}$), whereas a negatively charged GO membrane would show a tendency to exclude A$_2$B salt containing the divalent anion (B$^{2-}$). The permeation of salts appeared to be dominated by electrostatic repulsion of the charged membrane surface against high-valent co-ions, although there was also an electrostatic attraction between membrane surface charges and a low-valent counter-ion. Accordingly, we speculated that ion transport through the surface-charged GO membrane is also related to the ion valence, which might determine the electrostatic interactions with the membrane surface as well. We therefore conducted a set of filtrations by varying the valence ratio of cation and anion ($Z^+/Z^-$) of the salts (Fig. 2c, d). The results indicated that salts permeation through the surface-charged GO membranes was closely dependent on the valence ratio of cation and anion. For membrane surfaces with positive charges (e.g., GO-PDDA) that repelled the cation while attracting the anion, salt permeation was suppressed as the $Z^+/Z^-$ changed from 2/1 to 1/2, leading to water/salt selectivity ranked in the order of MgCl$_2$ > MgSO$_4$ > Na$_2$SO$_4$. In contrast, the relationship between salt permeability and water/salt selectivity versus $Z^+/Z^-$ displayed the reverse order in negatively charged membranes (e.g., GO-PSS) where the cation was attracted while the anion was repelled. There is a competition between electrostatic repulsion of co-ions against the membrane surface and the electrostatic attraction of counter-ions toward the membrane surface. In case of either $Z^+/Z^- < 1$ (e.g., Na$_2$SO$_4$) for positively charged membranes or $Z^+/Z^- > 1$ (e.g., MgCl$_2$) for negatively charged membranes, the attraction of high-valent counter-ions could dominate over the repulsion of low-valent co-ions, thereby facilitating the salt permeation through the membrane.

Interestingly, the NaCl transport behaviors were almost unchanged with positively and negatively charged GO membrane (Supplementary Fig. 11), owing to a balanced electrostatic interaction with monovalent co-ions and counter-ions. The variation of MgSO$_4$ permeability in positively and negatively charged GO membranes reflects the additional size discrimination effect on the ionic transport (detailed discussion can be found in the note for Supplementary Fig. 11). In addition to the salt valence ratio that controls ion transport based on the dominant electrostatic exclusion effect in dilute salt solution, the salt concentration is another factor affecting ion transport via an electrostatic screening effect. MgCl$_2$ (or Na$_2$SO$_4$) permeability of the positively (or negatively) charged GO membrane was enhanced by increasing the salt concentration in the feed solution (Supplementary Fig. 12). A GO membrane with a given surface charge density possesses a certain capacity of repelling or attracting ions. Once excessive ions have been introduced, e.g., feeding with a high salt concentration, the charge screening effect would lessen the exclusion effect, which ultimately contributes to the higher ion transport rate through the membrane.

**Ion transport mechanism through surface-charged GO membrane.** To support our hypothesis, we further examined the underlying ion transport mechanism taking place in the surface-charged GO membranes with the aid of theoretical arithmetic.

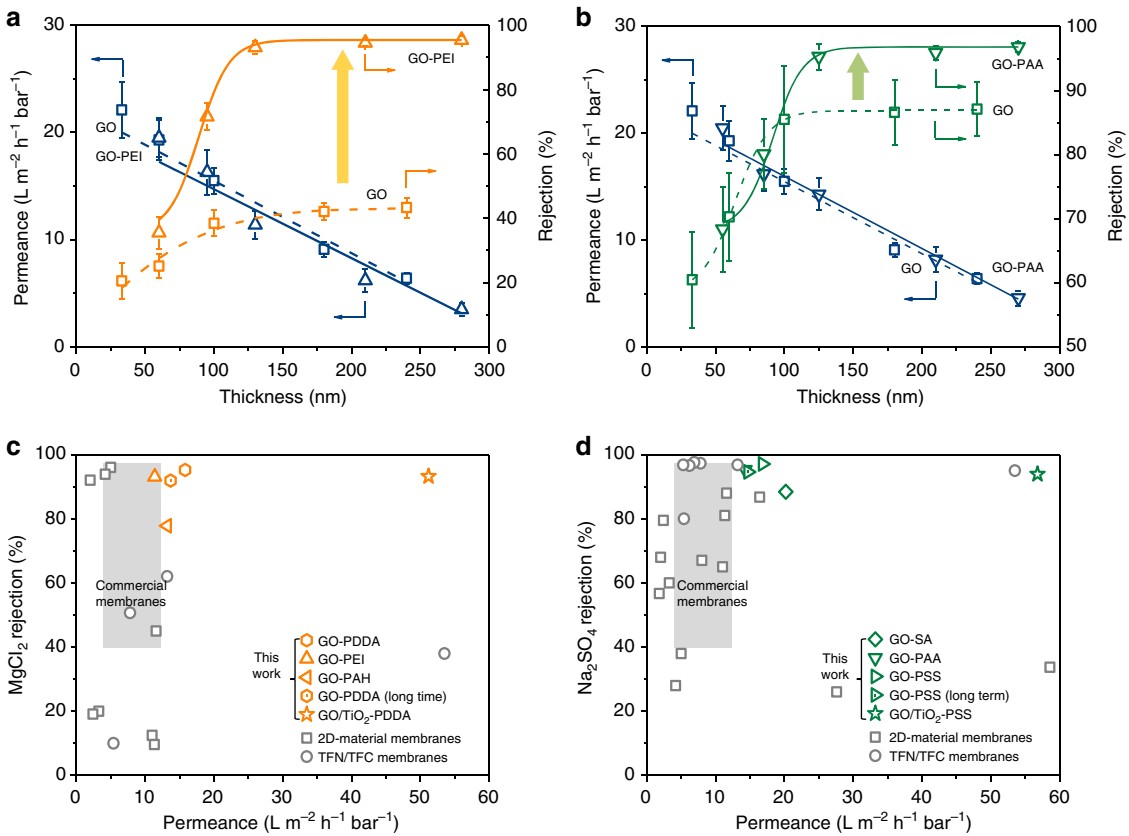

**Fig. 3** Membrane performance comparison. **a** Water permeance and MgCl$_2$ rejection of pristine graphene oxide (GO) and positively charged GO-PEI membrane as a function of membrane thickness (GO deposition amounts of 0.1, 0.2, 0.5, 0.8, and 1.0 mg with 0.1 wt% polyethylene imine (PEI) surface coating) under 2 bar filtration at feed concentration of 50 ppm. **b** Water permeance and Na$_2$SO$_4$ rejection of pristine GO and negatively charged GO-PAA membrane as a function of membrane thickness (GO deposition amounts of 0.1, 0.2, 0.5, 0.8, and 1.0 mg with 0.1 wt% polyacrylic acid (PAA) surface coating) under 2 bar filtration at feed concentration of 50 ppm. Dashed lines: pristine GO membranes; solid lines: surface-charged GO membranes. Yellow and green upward arrows indicate the remarkable improvements of surface-charged GO membranes in salt rejection. Error bars represent standard deviations for three measurements. **c** MgCl$_2$ rejection with water permeance of positively charged GO membranes (GO-PDDA marked as orange hexagon, GO-PEI marked as orange up-triangle, GO-PAH marked as orange left-triangle, GO-PDDA in long time measurement marked as orange spotted hexagon, TiO$_2$ intercalated GO-PDDA marked as orange star). **d** Na$_2$SO$_4$ rejection with water permeance of negatively charged GO membranes (GO-SA marked as green hexagon, GO-PAA marked as green up-triangle, GO-PSS marked as green left-triangle, GO-PSS in long time measurement marked as green spotted hexagon, TiO$_2$ intercalated GO-PSS marked as green star) in this work, as well as comparison with two-dimensional-material membranes (marked as gray squares), thin film nanocomposite (TFN) and/or thin film composite (TFC) membranes (marked as gray circles), and commercial polymeric nanofiltration membranes (marked as gray regions, e.g., NF270, NF90, NF200 membranes from Dow; DK, DL series of membranes from GE; ESNA series of membranes from Hydranautics). For references, see Supplementary Tables 2 and 3 in detail

The repulsive force of the charged membrane surface against co-ions can be reflected by the interaction energy between ions and the membrane surface (Fig. 2e and Supplementary Fig. 13), which was calculated by using the Derjaguin–Landau–Verwey–Overbeek (DLVO) theory[24,25] that involves estimation of Van der Waals and electrostatic double-layer interaction energies. Van der Waals attraction was determined using Hamaker's microscopic approach, and electrostatic repulsion was derived from the solution of Poisson–Boltzmann equation. We employed a surface element integration (SEI) model for the DLVO interaction calculation (Fig. 2f), which considers the total interaction energy between an ion and the membrane surface by integrating the interaction energy per unit area. We compared the net DLVO interaction energy curves (Fig. 2e–g) for (i) the highly positively charged GO-PDDA membrane against divalent co-ions Mg$^{2+}$ versus monovalent Na$^+$; (ii) the highly negatively charged GO-PSS membrane against divalent co-ions SO$_4^{2-}$ versus monovalent Cl$^-$; and (iii) the highly negatively charged GO-PSS membrane versus the negatively charged pristine GO membrane against SO$_4^{2-}$. Surprisingly, the charged membrane surface exhibited a

great interaction energy barrier for high-valent co-ion transport that exponentially decayed for the low-valent co-ion either in the positive charge case (i) or the negatively charge case (ii). It is reasonable to assume that the attraction of the charged membrane surface for the counter-ions would follow the same rule. Thus these results demonstrate the possibility of controlling the ion transport through a designed surface-charged membrane aimed at a given type of salt. For example, for a given AB$_2$-type salt, a positively charged membrane would be expected to exhibit a much higher force in repulsion to A$^{2+}$ than in attraction to B$^-$, thereby tending to repel A$^{2+}$ co-ions from the membrane. To balance the charge in solution, B$^-$ counter-ions would be simultaneously excluded. This accounts for the fact that the observed AB$_2$-type salt (e.g., MgCl$_2$) permeability is several folds lower than that of A$_2$B-type salt (e.g., Na$_2$SO$_4$) in positively charged GO membranes. Similarly, A$_2$B-type salt (e.g., Na$_2$SO$_4$) permeation is restricted by negatively charged GO membranes. In addition, apparently, increasing the interaction energy barrier by enhancing the surface charge density in case (iii) is efficient to improve the exclusion toward salts containing divalent co-ions.

Therefore, a membrane with highly positive charges exhibits a restraint of the transport of MgCl$_2$, whereas a membrane with highly negative charges generates a significant energy barrier for the transport of Na$_2$SO$_4$.

**Nanofiltration performance of surface-charged GO membrane.** The controllable ion transport achieved in surface-charged GO membranes encouraged us to apply them in nanofiltration. The charged GO membranes with highly tunable exclusion against divalent salts while allowing free permeation of monovalent salts (Supplementary Fig. 14) perfectly fit the spectrum of the nanofiltration process, which is used to remove polyvalent salts while reserving beneficial mineral salts in applications such as the production of potable water[5,6]. Nanofiltration is also regarded as a loose-structure and low-pressure alternative to reverse osmosis for high-throughput and low-energy desalination applications[26]. We measured the separation performance of pristine GO and surface-charged GO membranes in the nanofiltration process (Fig. 3a, b). The pristine GO membranes allowed fast water transport but failed to sieve salts out of water with a sharp permeation cut-off of hydrated radii ~4.7 Å (Supplementary Fig. 15), which was determined by the intrinsic interlayer distance for GO laminates that swell in water[16,21,27]. In real water treatment applications, appropriate permeability (water permeance) as well as enhanced selectivity (salt rejection) are critically required for high efficiency of desalination processes[26]. Remarkable improvement in salts rejection was stimulated by altering the surface charges of the GO membrane. The positively charged GO-PEI membrane exhibited MgCl$_2$ rejection up to ~95%, which is 2.3 times higher than that of the optimized GO membrane (~42%). Similarly, the highest Na$_2$SO$_4$ rejection of ~86% for the pristine GO membrane could be further increased to ~96% in the negatively charged GO-PAA membrane. Noting that a critical thickness of GO membrane (~100 nm in our case) is needed to provide an ultra-smooth and defect-free platform for the uniform deposition of polyelectrolyte layer, and thus well-distributed surface charges can be achieved to perform effective electrostatic exclusion function for the membrane. Notably, the water permeance had almost no decrease in these surface-charged GO membranes, indicating that fast water permeation channels within the GO laminates are well preserved. By contrast, the use of conventional hybrid approaches, such as layer-by-layer or mixed matrix to incorporate polyelectrolytes into GO laminates, severely compromised these fast water transport channels, resulting in 4–5 times lower water permeance than the pristine GO membrane with similar membrane thickness (Supplementary Figs 16–18).

Based on the attractive salts exclusion capability derived from surface charges of the GO membrane, we further improved the water permeance (from ~15 to ~56 L m$^{-2}$ h$^{-1}$ bar$^{-1}$) without substantially reducing salts rejection by intercalating nanoparticles into the surface-charged GO laminate (Supplementary Figures 19 and 20)[28,29]. This promising result suggests that the surface charge control approach could also allow an independent optimization of GO nanochannels to further boost water transport. In addition, the attachment of polyelectrolytes allows regulation of the surface hydrophilicity of the GO membrane to better tune the sorption behavior of water or other components in the feed (Supplementary Fig. 21), which can contribute to accelerated water permeation (Supplementary Fig. 22). Distinct from existing GO-based membranes with carefully tuned transport channels that often suffer a trade-off between salt rejection and water permeance, our surface-charged GO membrane have achieved remarkable advancement in salts rejection without compromising fast water permeance. The rationally designed surface-charged GO membranes exhibit MgCl$_2$ rejection of 93.2% with water permeance of 51.2 L m$^{-2}$ h$^{-1}$ bar$^{-1}$ and

Na$_2$SO$_4$ rejection of 93.9% with water permeance of 56.8 L m$^{-2}$ h$^{-1}$ bar$^{-1}$, which are far beyond the performance limits of GO membranes (Fig. 3c, d). Such excellent performance is superior to that of most state-of-the-art nanofiltration membranes including two-dimensional (2D)-material membranes (marked as squares), thin film nanocomposite and/or thin film composite membranes (marked as circles), and commercial polymeric nanofiltration membranes (marked as gray regions) (Tables S1 and S2). Also, the surface-charged GO membranes with controllable ion transport properties clearly overcome the salt permeability and water/salt selectivity trade-off observed for many polymeric membranes (Supplementary Fig. 23)[30].

We also employed our surface-charged GO membranes under aggressive high-pressure and long-term operation conditions that reflect the practical stability and feasibility of these membranes. Promisingly, we observed that the separation performance of our surface-charged GO membranes remained almost stable over the high pressure of 6 bar and the long period of 120 h (Supplementary Fig. 24). Moreover, we demonstrated scalability of the facile surface charge controlling strategy by fabricating a 15 cm-diameter surface-charged GO membrane via the same approach whose effective area (176.7 cm$^2$) is 10–60 times larger than the reported GO-based membranes (Supplementary Table 4). Four small pieces of membranes incised from different locations of the large membrane exhibited desirable salt retention capability with water permeance of ~10.5 L m$^{-2}$ h$^{-1}$ bar$^{-1}$ and MgCl$_2$ rejection of ~90% (Supplementary Fig. 25).

In summary, our work demonstrates a methodology for manipulation of surface charge to realize controllable ion transport through a GO membrane, in which desirable electrostatic interactions with charged ions were successfully created and finely tailored by the attachment of ionizable functional groups with various protonation/deprotonation abilities on the surface of pre-stacked GO laminates. The proposed ion transport mechanism clarified the controlling factors of the interaction energy barrier that arose between charged ions and the charged membrane surface. Surface polyelectrolyte layer with tunable charge properties offered desirable interactions with charged ions to control the ionic transport, meanwhile underlaying GO laminate with 2D graphene capillaries provided fast water transport nanochannels. By rational design of the membrane surface charge and the transport channels, the resulting surface-charged GO membranes exhibited outstanding rejection of salts and ultrahigh water permeance in a nanofiltration process, whose performance was far beyond the performance limit of state-of-the-art nanofiltration membranes. The approach of tuning surface charges to control ion transport, demonstrated here, establishes a platform that could be of interest in a variety of applications, such as water transport, studies of biomimetic ion channels and biosensors, ion batteries, and energy conversions.

## Methods

**Membrane preparations.** GO powder was prepared by modified Hummer's method and then was dissolved in deionized water followed by ultrasonication for 30 min to obtain stable and homogeneous GO aqueous suspension. GO membranes were prepared by a pressure-assisted filtration-deposition method. The sheet-flat PAN ultrafiltration membrane with a molecular weight cut-off of 100,000 Da was used as substrate. GO nanosheets in the aqueous suspension uniformly deposit on the substrate to form well-assembled GO laminates under the pressure of 2 bar using a self-designed filtration device. The GO membranes with different thickness were obtained by depositing different amounts (0.1, 0.2, 0.5, 0.8 and 1.0 mg) of GO nanosheets.

The surface-charged GO membranes were further prepared by a simple dip-coating method. PDDA, PEI, PAH, PSS, PAA, and SA are selected on basis of their different charge intensity and water sorption ability. The polyelectrolyte coating concentrations can be easily tuned (0.05, 0.1, 0.2, 0.3 wt%). Specifically, 0.1 wt% polyelectrolyte aqueous solutions were poured into the device and kept still for 30 min before being poured out. The final membranes were rinsed with deionized water and

dried at room temperature. The resultant membranes were referred to as GO-PDDA, GO-PEI, GO-PAH, GO-PSS, GO-PAA, and GO-SA membranes, respectively.

**Membrane characterizations.** The membrane surface morphologies and membrane thicknesses were imaged and measured by field-emission scanning electron microscope (S4800, Hitachi, Japan) at a voltage of 5 kV and current of 10 μA. The membranes surface phase and height profiles with roughness data were measured by atomic force microscopy (XE-100 Park SYSTEMS, Korea) in the range of 5 × 5 μm² operated in the non-contact mode.

The zeta potential of polyelectrolyte aqueous solutions (0.1 mg mL⁻¹, pH 7) were investigated by Zeta potential analyzer (Zetasizer Nano ZS90, Malvern, UK). The surface charge properties of membranes were analyzed by a SurPASS electrokinetic analyzer (Anton Paar GmbH, Austria) through streaming potential measurements. A 0.001 M KCl solution was used to measure the zeta potential of the membrane initially under neutral pH. After that, the pH of the solution increased gradually to pH 11 and then decreased to pH 2.6 by autotitration with 0.1 M HCl and 0.1 M NaOH solutions, respectively.

Fourier transform IR (AVATAR-FT-IR-360, Thermo Nicolet, USA) spectra of membranes in the range of 4000–750 cm⁻¹ were displayed to characterized the surface functionalized groups. XPS (Thermo ESCALAB 250, USA) was employed to determine the surface chemistry of membranes. The crystal phases of the samples were examined by X-ray diffraction (model D8 Advance, Bruker) with Cu Kα radiation to further calculate the $d$-spacing of the membranes.

A spectroscopic ellipsometer (Compete EASEM-2000 U, J. A. Woollam) with the wavelength ranged from 250 to 1000 nm at an incident angle of 70° was applied to measure the thickness of the polyelectrolyte layers coated on the surface of Si wafers. The B spline model was used to fit data. More than four spots on the surface of Si wafer were selected and the average value was reported for the thickness.

The quartz crystal microbalance technique (QCM200 Quartz Crystal Microbalance, Stanford Research Systems, Inc.) was used to evaluate the water sorption ability of polyelectrolytes. The polyelectrolyte layer was coated onto a gold-coated quartz sensor, and air with certain humidity was driven into the QCM chamber with dynamic weight data recorded. The surface hydrophilicity of membranes was evaluated by detecting the static water contact angle at room temperature using a contact angle measurement system (Drop Shape Analyzer-DSA100, Kruss, Germany).

Salts concentrations in feed and permeate solutions were obtained using electrical conductivity (FE38-Standard, METTLER TOLEDO, Switzerland).

**Calculation of membrane surface charge density.** The membrane charge density ($\sigma$, mC m⁻²) is calculated by membrane zeta potential according to Gouy–Chapman equation[31] as follows:

$$\sigma = -\epsilon\kappa\xi \frac{\sinh\left(\frac{F\xi}{2RT}\right)}{\frac{F\xi}{2RT}} \tag{1}$$

where $\kappa^{-1} = \left(\frac{\epsilon RT}{2F^2 C}\right)^{\frac{1}{2}}$ is Debye length, $\xi$ (mV) is membrane zeta potential obtained through streaming potential measurements, $R = 8.3145$ J mol⁻¹ K⁻¹ is gas constant, $F = 96485$ C mol⁻¹ is Faraday constant, $T = 298$ K is absolute temperature, and $\epsilon = 6.933 \times 10^{-10}$ F m⁻¹ is permittivity.

**Membrane performance measurements.** Membrane performance is tested using nanofiltration process under 2 bar by a self-designed filtration device at room temperature. The effective area of the membranes is 17.35 cm². The water permeance ($J$, L m⁻² h⁻¹ bar⁻¹) was measured with deionized water, and the salt rejection ($R$, %) was determined using salts (i.e., MgCl₂, Na₂SO₄, NaCl) in the form of 50 ppm aqueous solutions. Each data was obtained by a new membrane sample, and at least three membranes were tested to validate the reproducibility. The membranes were first conditioned under nanofiltration operation for 2 h before collecting permeate samples. During the filtration process, we recycled the permeation into the feed tank to maintain a stable salt concentration in the feed.

Water permeance and salt rejection are calculated as follows:

$$J = \frac{V}{A \times t \times P} \tag{2}$$

$$R = \left(1 - \frac{c_p}{c_f}\right) \times 100\% \tag{3}$$

where $V$ is the volume of permeate collected (L), $A$ is the membrane effective area (m²), $t$ is the permeation time (h), $P$ is the applied pressure (bar), and $c_p$ and $c_f$ are the concentrations of the permeate and feed solution, respectively.

The transport of water and salt through membranes was described in terms of the solution–diffusion model[23]. The water flux, $J_W$ (g cm⁻² s⁻¹), can be expressed as follows:

$$J_W = \frac{D_W}{L}\left(C_{W,F}^m - C_{W,P}^m\right) = \frac{D_W C_{W,F}^m}{L}\left(1 - \frac{C_{W,P}^m}{C_{W,F}^m}\right) \tag{4}$$

where $D_W$ (cm² s⁻¹) is the average water diffusion coefficient in the membrane; $L$

(cm) is the thickness of the membrane; and $C_{W,F}^m$ and $C_{W,P}^m$ (g cm⁻³) are water concentration in the membrane on the feed and permeate side, respectively. On account of the pressure and osmotic pressure difference between the feed and permeate sides of the membrane, Eq. (4) is often written as follows:

$$J_W = \frac{D_W C_{W,F}^m}{L}\frac{\bar{V}}{RT}(\Delta P - \Delta\pi) \tag{5}$$

where $\bar{V}$ (cm³ mol⁻¹) is the partial molar volume of water, which is typically well approximated by the molar volume of pure water when the water uptake varies little with salt concentration over the salt concentration range of interest; $R$ (83.1 cm³ bar mol⁻¹ K⁻¹) is the gas constant, $T$ (K) is the absolute temperature; $\Delta P$ (bar) is the pressure difference across the membrane, and $\Delta\pi$ (bar) is the osmotic pressure difference across the membrane.

The water partition (or solubility) coefficient, $K_W$, is defined as the ratio of water concentration in the membrane to that in the contiguous solution:

$$K_W = \frac{C_{W,F}^m}{C_{W,F}} \tag{6}$$

For relatively dilute solutions, $C_{W,F}$ is approximately equal to the density of pure water, $\rho_W$ (g cm⁻³). Combining Eqs. (5) and (6) yields:

$$J_W = \frac{P_W}{L}\frac{\rho_W \bar{V}}{RT}(\Delta P - \Delta\pi) = \frac{P_W}{L}\frac{M_W}{RT}(\Delta P - \Delta\pi) = A(\Delta P - \Delta\pi) \tag{7}$$

where $A$ is the effective membrane permeance to water, and $P_W$ is the membrane permeability to water. As indicated in Eq. (7), $A$ is related to water permeability $P_W$ as follows:

$$A = \frac{P_W}{L}\frac{M_W}{RT} \tag{8}$$

According to the solution–diffusion model, the salt flux through membrane, $J_S$ (g cm⁻² s⁻¹), can be given as:

$$J_S = \frac{P_S}{L}\left(C_{S,F} - C_{S,P}\right) = \frac{P_S}{L}\Delta C_S = B\Delta C_S \tag{9}$$

where $P_S$ (cm² s⁻¹) is the salt permeability; $C_{S,F}$ and $C_{S,P}$ (g cm⁻³) are the salt concentrations in the solution on the feed and permeate sides of the membrane, respectively, and $\Delta C_S$ is the salt concentration difference. It worth noticing that our focus is on salt and water transport properties through the membrane, so the concentration polarization is not considered. In addition, $B = P_S/L$ is the reported salt flux. $\Delta C_S$ and $\Delta\pi$ are typically related as $\Delta\pi = \Delta C_S RT$.

The capability of the membrane to remove salt from a feed solution is often characterized in terms of salt rejection, $R$ (%), which can be presented as follows within the context of the solution–diffusion model:

$$R = \frac{(P_W/P_S)(\bar{V}/PT)(\Delta P - \Delta\pi)}{1 + (P_W/P_S)(\bar{V}/PT)(\Delta P - \Delta\pi)} \times 100\% = \frac{(A/B)(\Delta P - \Delta\pi)}{1 + (A/B)(\Delta P - \Delta\pi)} \times 100\% \tag{10}$$

Consequently, water permeability and salt permeability can be expressed as follows:

$$P_W = K_W D_W \tag{11}$$

$$P_S = K_S D_S \tag{12}$$

The ideal water/salt selectivity, $\alpha_{W/S}$, is defined as the ratio of water permeability to salt permeability:

$$\alpha_{W/S} = \frac{P_W}{P_S} = \frac{K_W}{K_S} \times \frac{D_W}{D_S} \tag{13}$$

where $K_W/K_S$ is the water/salt solubility selectivity, and $D_W/D_S$ is the water/salt diffusivity selectivity.

**Calculation of DLVO interaction.** The DLVO interactions between charged ions and surface-charged GO membranes are calculated using the SEI technique[24]. The SEI technique considers the total interaction energy between hydrated ion and planar membrane surface by integrating the interaction energy per unit area:

$$U(D) = \int\int E(h)dA \tag{14}$$

Here $U$ is the interaction energy between hydrated ion and membrane surface, $D$ is the closest distance between them, $E$ is the interaction energy per unit area between ion and membrane surface separated by a distance $h$, and $dA$ is the projected differential surface area of the ion.

To provide a facile description of the mathematical formulation, the analysis presented here employs a cylindrical coordinate system, and the expression for the interaction energy becomes:

$$U(D) = \int_0^{2\pi}\int_0^a E(h)r\,dr\,d\theta \tag{15}$$

$$h = D + a - \sqrt{a^2 - r^2} \tag{16}$$

where $a$ is radius of the hydrated ion and $h$ is the vertical distance between a circular arc (differential surface area $r\, dr\, d\theta$) of hydrated ion and the point on the membrane surface directly below it.

In this study, we use the DLVO interaction energy per unit area between hydrated ion and membrane surface obtained by adding the Hamaker expression for the van der Waals interaction and the constant potential electrostatic double-layer interaction energy expression. The total DLVO interaction energy per unit area is thus given as:

$$E_{DLVO}(h) = E_{VDW}(h) + E_{EDL}(h) = -\frac{A_H}{12\pi h^2}$$
$$+ \frac{\epsilon\epsilon_0\kappa}{2}\left[\left(\psi_s^2 + \psi_m^2\right)\left(1 - \coth\kappa h\right) + \frac{2\psi_s\psi_m}{\sinh\kappa h}\right] \quad (17)$$

Here $A_H$ is Hamaker constant, $\epsilon$ is dielectric constant of solvent, $\epsilon_0$ is dielectric permittivity of vacuum, $\psi_s$ is surface potential of hydrated ions, $\psi_m$ is surface potential of surface-charged GO membrane, and $\kappa$ is the inverse Debye screening length, respectively.

The surface potential of hydrated ions is calculated according to Coulomb's law as follows:

$$\psi_s = \frac{q}{4\pi\epsilon r} \quad (18)$$

where $\psi_s$ is surface potential of hydrated ion, $q$ is the charge of the ion (which equals to $1.602 \times 10^{-19}$ C for monovalent ions such as Na$^+$ and Cl$^-$, and $3.204 \times 10^{-19}$ C for divalent ions such as Mg$^{2+}$ and SO$_4^{2-}$), $\epsilon$ is dielectric constant, and $r$ is the hydration radius of hydrated ions (which is 0.358, 0.332, 0.428, and 0.379 nm for Na$^+$, Cl$^-$, Mg$^{2+}$, and SO$_4^{2-}$, respectively[32]).

## Data availability
The source data underlying Figs. 1d, e, 2a–g, and 3a–d are provided as a Source Data file. The data that support the findings of this study are available from the corresponding author upon reasonable request.

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

## Acknowledgements
This work was financially supported by the National Natural Science Foundation of China (grant nos. 21490585, 21776125, and 51861135203), the Innovative Research Team Program by the Ministry of Education of China (grant no. IRT17R54), and the Topnotch Academic Programs Project of Jiangsu Higher Education Institutions (TAPP). We thank Shipeng Sun for offering commercial NF membrane samples and Andrew Livingston, Bill Koros and Ho Bum Park for helpful discussions.

## Author contributions
W.J. and G.L. conceived the idea. W.J., G.L. and M.Z. designed the experiments and wrote the manuscript. M.Z. performed the experiments. M.Z., K.G. and Y.J. prepared the data graphs. M.Z., K.G., Y.J., G.L., W.J. and N.X. discussed the results and commented on the manuscript.

## Additional information

**Competing interests:** The authors declare no competing interests.

