## [Peer Review File · Nature Communications]

Reviewers' comments:

Reviewer #1 (Remarks to the Author):

Comments

In this paper, the authors try to manipulation of the membrane surface charge to control ions transport without impeding fast water permeation through GO membrane. Due to the electrostatic repulsion, the resulting membranes showed the selective ion transport property. The work is also well summarized the deliberately regulated surface charged of GO and modified on the surface of nanofiltration membranes to selective separation ions. However, some details and questions should be clarified before possible publication. A revision is suggested.

1. Page 1, line 34. Selective ion transport has been observed in GO membranes based on computer simulations and self-diffusion measurements,

What is "Selective ion"? The authors should demonstrate clearly. Charge, size of ions or others ?

2. Page 1, line 35. but the practicality of these membranes has yet to be demonstrated.

However, lots of paper reported that they have demonstrated a GO based membrane with selective separation of monovalent ions.

3. Page 2, line 39. Numerous attempts to precisely modulate the interlayer spacing of GO membranes have been undertaken.

There are also some reports about preparation of 3D GO membranes with tunable nanoscale interlayer, such as *J. Mater. Chem. A*, 2018, 6, 18859-18864; *Adv. Mater. Interfaces* 2018, 5, 1701449.

4. Page 2, Figure 1. Design of surface-charged GO membrane.

What's the charge of GO nanosheet? Why positively or negatively charged polymers can be coated on GO surface? Is it due to the electrostatic adsorption, Van der Waals force or formed a chemical bond?

5. Page 3, line 45. First, we prepared well-assembled GO laminates on top of a porous polyacrylonitrile (PAN) substrate, followed by the attachment of tunable surface charges via dip-coating an array of selected polyelectrolytes (polycations: PDDA, PEI, PAH; polyanions: PSS, PAA, SA) on the surface of pre-stacked GO laminates to create surface-charged GO membranes (Fig. 1b and Supplementary Fig. S1).

That means the polyelectrolytes only coated on the top of GO layers. How to control the stability of GO nanosheets? Are the GO nanosheets easy separate each other in aqueous? Why?

Reviewer #2 (Remarks to the Author):

The manuscript by Zhang et al. is interesting based on the simple procedure adopted to achieve surface charge control on graphene oxide (GO) via dip coating suitable polyelectrolytes and its stability over 120 hrs shown in the dead-end filtration set up. Even though the concept of surface charge control of GO membranes are not new, the reported results are clear and convincing. The data

presented are consistent with the claims and supported by a qualitative model. The role of surface charge on the salt repulsion (at low feed concentration) as shown in fig 2 is convincing. Overall, the article is well written and could be publishable in Nature Communications after clarifying the following concerns.

1) The polyelectrolyte coating has shown to reduce the surface roughness of GO significantly.

What was the physical thickness of the polyelectrolyte layer? It is shown that Permeance is not much affected by the wt% of the polyelectrolyte coating. What is the justification for this?

2) Based on the interaction energy presented in fig 2e, how to justify the observed feed concentration dependence on the salt rejection/selectivity?

3) What was the feed concentration used in Fig 3 a, b?

4) How is the surface potential of hydrated ions approximated in the model?

5) Identical morphology is shown in Fig S15 for surface charged GO and layer by layer assemble sample. What was the fundamental performance enhancing factor for the surface charged GO layers over the Layer-by layer GO-polyelectrolyte membranes?

6) Clarify whether the performance comparison with commercial membranes was performed at identical conditions including the feed concentration, pressure, etc.?

7) Define the solid line in Fig. 3c.

8) Need to justify the role of GO on the observed surface charge tuning or salt rejection properties of the studied membranes. Is the active layer GO or the polyelectrolyte?

Reviewer #3 (Remarks to the Author):

The article discusses the fabrication of graphene oxide membranes with tunable surface charge. By using different charged polyelectrolytes, the rejection in the GO membrane for divalent ions is tuned. The study is thorough, but the modification presented and the associated modeling are simple and similar methods have been performed elsewhere. Additionally, the observed impacts of surface charge on performance are predictable and well-understood. Further comments are given below.

It is well understood that surface charge in membrane will impact the rejection of charged solutes and that this effect will be reduced at higher concentrations. Additionally, several previous papers have used polyelectrolytes to create ion-selective nanofiltration membranes, including previous work with graphene oxide (<https://doi.org/10.1016/j.memsci.2014.06.036>). In this study, it is unclear how the graphene oxide base layer contributes uniquely to charge-based rejection that is observed, since the coating only impacts the surface and does not exploit the confinement between GO sheets.

Figure 3: The trade-off curves drawn here appear to be arbitrary. The conventional trade-off for polymeric membranes is drawn with permeability of the solvent and solute on different axes; this method is useful because permeability is close to an intrinsic material property. As it is currently

drawn in Fig 3, the curve does not appear to have much fundamental basis. Why should the trade-off be a linear curve with rejection on the y-axis? Also, comparing solely based on rejection is not justifiable since the effect of salt concentration and operating conditions is not accounted for.

Line 33: The frictionless flow in graphene oxide membranes has been disputed by a number of studies (DOI: 10.1021/acs.nano.8b02015).

Reviewer #4 (Remarks to the Author):

The rejection of GO membranes for ions under pressure remains a big challenge. In this work, the authors achieved an exceptional enhancement of ion rejection without compromising GO membranes' water permeance by manipulating the membrane surface charge properties. The idea is great and experiments carried out are comprehensive. Still, a few questions.

1. In page 3, line 67-70, the photograph shows a large area GO membrane (diameter 15 cm) with GO loading amount of 5 mg. The SEM image is for a GO membrane with GO loading amount of 0.5 mg. I guess the diameter should be 4.7 cm. Also, the captions for Fig. S2 should be modified to avoid confusion.

2. Is the solution-diffusion model suitable for GO membranes? It is widely used for RO membranes. For NF membranes where electrostatic interaction is an important consideration, the Donnan equilibrium model and the extended Nernst-Planck model may work better. What's more, GO membranes differ a lot from polymeric NF membranes. The nanocapillaries are horizontal, not vertical. These models also work in this case?

3. The authors found that the NaCl transport behaviors were almost unchanged with positively and negatively charged GO membranes because Z^+/Z^- is 1 for NaCl. Can any explanations be provided for the difference in the permeability of MgSO₄?

4. It can be concluded from the authors' findings that the surface charge properties of GO membranes dominate their ion rejection performance. Fig. 3(a-b) shows that GO membranes' rejections for MgCl₂ and Na₂SO₄ are low when their thicknesses are smaller than 125 nm, even if they are charged. Can any possible reasons be provided?

5. I'm interested in the long period testing of the membranes. Assuming that the testing was carried out under 1 bar, water flux of the membranes as 15 LMH/bar, effective filtration area as 17.35 cm², the volume of the filtrate should be more than 3 L. A huge feed container was used? Or, the testing was carried out using a cross-flow system?

Responses to Reviewers' Comments

Manuscript #: NCOMMS-18-33220-T

Title: Controllable ion transport by surface-charged graphene oxide membrane

We highly appreciate the reviewers for their detailed and instructive suggestions. Thanks so much for their valuable time and efforts. Their suggestions are really helpful for improving the quality of our present work. We agree with their advice and suggestions. We have done our best to comply with them in the revised manuscript. The following are our responses to their comments one by one.

Reviewer #1

General Comment

In this paper, the authors try to manipulation of the membrane surface charge to control ions transport without impeding fast water permeation through GO membrane. Due to the electrostatic repulsion, the resulting membranes showed the selective ion transport property. The work is also well summarized the deliberately regulated surface charged of GO and modified on the surface of nanofiltration membranes to selective separation ions. However, some details and questions should be clarified before possible publication. A revision is suggested.

Response to General Comment

We are grateful for the positive evaluation of our work. All the revised parts can be found in the revised manuscript with changes marked.

Comment 1

Page 1, line 34. Selective ion transport has been observed in GO membranes based on computer simulations and self-diffusion measurements. What is “Selective ion”? The authors should demonstrate clearly. Charge, size of ions or others?

Response to Comment 1

Thanks for reviewer’s suggestion. “Selective ion transport in GO membrane” is for the expression of the phenomenon that some specific ions can permeate through GO membrane whereas other ions would be rejected by the GO membrane. We reviewed three representative studies involving selective ion transport in GO membranes mainly based on size sieving effect and interaction between ions and GO membrane shown as below:

Ref. 10: Grossman et al. used classical molecular dynamics to predict the salt ions passage depending critically on pore diameter of graphene membrane (*J. C. Grossman et al., Nano Lett. 12, 3602–3608, 2012*).

Ref. 11: Nair et al. observed a sharp cut-off at ~ 4.5 Å in GO membrane which blocks all solutes with hydrated radii larger than 4.5 Å while allowing small ions

permeate through GO membrane quickly (*A.K. Geim et al., Science 343, 752-754, 2014*).

Ref. 12: Zhu et al. demonstrated sodium salts permeate quickly through GO membrane whereas heavy-metal salts infiltrate much more slowly, attributing to the coordination between heavy-metal ions with GO membrane restricting their passage (*H. Zhu et al., ACS Nano 7, 428-437, 2012*).

In order to make it more clearly, we have revised the statement (Line 37, Page 2) in the manuscript as follow:

“It has been reported that specific ions could selectively permeate through GO membrane, achieving from computer simulations¹⁰ and self-diffusion measurements^{11,12}, mainly based on size sieving effect and interaction between ions and GO membrane.”

Comment 2

Page 1, line 35. But the practicality of these membranes has yet to be demonstrated.

However, lots of paper reported that they have demonstrated GO based membrane with selective separation of monovalent ions.

Response to Comment 2

Thanks for the comment. Here these membranes are referred to pristine GO membranes (GO mixed-matrix membranes are not included here) showing selective ion transport based on computer simulations or self-diffusion measurements. The fabrication method and procedure of these membranes are relatively complicated and remain challenging to realize large-area and high-throughput manufacture. Additionally, the self-diffusion driven separation is not an industrially preferred process in terms of efficiency. To make the statement more clearly, we have revised it (Line 39, Page 2) in the manuscript as follow:

“While the high-throughput manufacture and industrial implementation of these GO membranes need to be further studied to demonstrate their practicality and scalability.”

Comment 3

Page 2, line 39. Numerous attempts to precisely modulate the interlayer spacing of GO membranes have been undertaken. There are also some reports about preparation of 3D GO membranes with tunable nanoscale interlayer, such as J. Mater. Chem. A, 2018, 6, 18859-18864; Adv. Mater. Interfaces 2018, 5, 1701449.

Response to Comment 3

Thanks for reviewer's suggestion. The suggested references are indeed very important studies for developing 3D GO membranes with tunable nanoscale interlayer. We have reviewed these key progresses (Line 44, Page 2) in the revised manuscript as below:

“Numerous attempts to modulate the interlayer spacing of GO membranes have been undertaken by partial reduction¹⁵, cross-linking¹⁶ and building multilayer architectures^{17,18}. Precise tuning GO interlayer spacing within subnanometer range is challenging¹⁹ and remains difficult to achieve high salt rejection in a pressured filtration process...”

Ref. 17. Y. Zhao et al. Formation of morphologically confined nanospaces via self-assembly of graphene and nanospheres for selective separation of lithium, *J. Mater. Chem. A* **6**, 18859-18864 (2018).

Ref. 18. Y. Zhao et al. Tunable nanoscale interlayer of graphene with symmetrical polyelectrolyte multilayer architecture for lithium extraction, *Adv. Mater. Interfaces* **5**, 1701449 (2018).

Comment 4

Page 2, Figure 1. Design of surface-charged GO membrane. What's the charge of GO nanosheet? Why positively or negatively charged polymers can be coated on GO surface? Is it due to the electrostatic adsorption, Van der Waals force or formed a chemical bond?

Response to Comment 4

Thanks for the reviewer's question. The strong molecular interactions between

GO and polyelectrolytes are the key to realize the design of surface-charged GO membrane, which mainly include hydrogen bonding and electrostatic adsorption. GO nanosheet is negatively charged due to the deprotonation of carboxyl and hydroxyl groups on GO. Benefiting from the abundant functional groups on polyelectrolytes (such as amino and sulfonic groups) and GO laminate (such as hydroxyl, carboxyl and epoxy groups), hydrogen bonding would be generated between polyelectrolyte layer and GO laminate through plentiful interaction sites. The shifts of characteristic peaks in FTIR spectra and XPS C 1s spectra of surface-charged GO membranes give evidence to the formation of hydrogen bonding among these functional groups on polyelectrolyte layer and GO laminate (Fig. S7-S8). In addition, according to the Zeta potential measurements (Fig. S10a), there is electrostatic attraction between positively charged polyelectrolyte and negatively charged GO nanosheets, further helping polyelectrolyte layer firmly attaching on GO laminate. We have added the explanation (Line 90, Page 4) in the manuscript as follow:

“The as-prepared surface-charged GO membrane preserved the laminar structure of the GO laminate (Supplementary Fig. S4) with an ultrathin polyelectrolyte layer firmly integrated on the surface via hydrogen binding and/or electrostatic attraction (Supplementary Fig. S7-S8, S10a).”

Fig. S7 (A) Schematic of probing interactions between polyelectrolyte layer and GO laminate. (B) Peak shifting in FTIR spectra of GO and surface-charged GO membranes.

Fig. S8 XPS C 1s and O 1s (N 1s or S 2p) spectra of (A) pristine GO and (B) surface-charged GO membranes.

Fig. S10 Zeta potentials of GO and polyelectrolyte aqueous solutions.

Comment 5

Page 3, line 45. First, we prepared well-assembled GO laminates on top of a porous polyacrylonitrile (PAN) substrate, followed by the attachment of tunable surface charges via dip-coating an array of selected polyelectrolytes (polycations: PDDA, PEI, PAH; polyanions: PSS, PAA, SA) on the surface of pre-stacked GO laminates to create surface-charged GO membranes (Fig. 1b and Supplementary Fig. S1). That means the polyelectrolytes only coated on the top of GO layers. How to control the stability of GO nanosheets? Are the GO nanosheets easy separate each other in aqueous? Why?

Response to Comment 5

Thanks for reviewer's question. The stability is indeed a critical issue for developing GO membrane. Our surface-charged GO membrane exhibited outstanding stability: the nanofiltration performance is remained stable over high pressure of 6 bar and continuous operation of 120 h, without involving defects or peeling off in the membrane. The stability of GO membrane is mainly affected by two aspects: GO layer swelling/re-dispersion and GO layer delaminating from the substrate. We thus control the stability of GO membrane from these two aspects. 1) For inhibiting GO layer swelling/redispersion: we selected GO material with a low O/C ratio of approximately 0.186 (15.68% O element, 84.32% C element) to minimize the adverse

effect of swelling. Our on-going work demonstrated the lower oxygenated GO nanosheets with larger hydrophobic regions could offer less action sites for hydration, which effectively restrains the intercalation and adsorption of water molecules (Fig. R1). Moreover, the surface coated polyelectrolyte layer could avoid the direct contact of GO membrane with massive water, inhibiting the re-dispersion of GO nanosheets in water. 2) For enhancing the adhesion between GO layer and substrate: we chose surface hydrolyzed PAN (h-PAN) substrate to provide robust interfacial adhesion with GO membrane according to our recent work on comparison of different substrates for preparation of GO membranes (*W. Jin et al., J. Membr. Sci. 574, 196-204, 2019*). The h-PAN substrate has abundant oxidized functional groups for remarkably enhancing hydrogen binding with GO laminate (Fig. R2).

Detailed membrane preparation description (Line 80, Page 4) has been added in the revised manuscript as follow:

“Firstly, we prepared **stabilized GO laminates using low O/C ratio (0.186) GO materials on top of a porous surface hydrolyzed polyacrylonitrile (h-PAN) substrate with strong interfacial adhesion²⁰**, followed by the attachment of tunable surface charges via dip-coating an array of selected polyelectrolytes (polycations: PDDA, PEI, PAH; polyanions: PSS, PAA, SA).”

Ref. 20: M. Zhang et al. Effect of substrate on formation and nanofiltration performance of graphene oxide membranes, *J. Membr. Sci.* **574**, 196-204 (2019).

Fig. R1 Schematic illustrations of the swelling of GO laminate prepared from (A) small lateral sized GO with high oxygenated degree and (B) large lateral sized GO

with low oxygenated degree.

Fig. R2 Schematic illustration of the assembly of GO laminates on different substrate based on hydrogen bond and Van der Waals force formed between functional groups on GO and substrate (*W. Jin et al., J. Membr. Sci. 574, 196-204, 2019*).

Reviewer #2

General Comment

The manuscript by Zhang et al. is interesting based on the simple procedure adopted to achieve surface charge control on graphene oxide (GO) via dip coating suitable polyelectrolytes and its stability over 120 hrs shown in the dead-end filtration set up. Even though the concept of surface charge control of GO membranes are not new, the reported results are clear and convincing. The data presented are consistent with the claims and supported by a qualitative model. The role of surface charge on the salt repulsion (at low feed concentration) as shown in Fig 2 is convincing. Overall, the article is well written and could be publishable in Nature Communications after clarifying the following concerns.

Response to General Comment

We highly appreciate the reviewer's positive comments. All the revised parts can be found in the revised manuscript with changes marked.

Comment 1

The polyelectrolyte coating has shown to reduce the surface roughness of GO significantly. What was the physical thickness of the polyelectrolyte layer? It is shown that Permeance is not much affected by the wt% of the polyelectrolyte coating. What is the justification for this?

Response to Comment 1

Thanks for reviewer's question. We have employed ellipsometer to measure the thickness of polyelectrolyte layer, indicating the physical thickness of polyelectrolyte layers ranging from 25 to 38 nm. Detailed conditions and results of ellipsometry measurement have been added in the Methods (Line 335, Page 12) and Supplementary Materials as follow:

“A spectroscopic ellipsometer (Compete EASEM-2000 U, J. A. Woollam) with the wavelength ranged from 250 to 1000 nm in an incident angle of 70° was applied to measure the thickness of the polyelectrolyte layers coated on the surface of Si

wafers. The B spline model was used to fit data. More than 4 spots on the surface of Si wafer were selected and the average value was reported for the thickness.”

Fig. S6 Schematic of ellipsometer for polyelectrolyte layer thickness measurement.

Table S1 Thickness of polyelectrolyte layers measured by ellipsometry.

Sample	PDDA	PEI	PAH	SA	PAA	PSS
Thickness (nm)	25±2	38±5	29±3	36±5	32±4	27±2

“Notes: Ellipsometry is an optical technique for investigating sample thickness. As illustrated in Fig. S6, electromagnetic radiation is emitted by a light source and linearly polarized by a polarizer, and then falls onto the sample. After reflection the radiation passes a modulator and a monochromator, and finally falls into the detector. The sample thickness can be calculated by further performing a model analysis. In order to obtain the physical thickness of polyelectrolyte layers, we prepared samples by coating polyelectrolytes on the surface of Si wafers using 0.1 wt% polyelectrolyte solutions. The results are summarized in Table S1.”

The limited influence of polyelectrolyte layer on the water permeance of surface-charged GO membranes can be attributed to two factors. 1) the attachment of ultrathin and loose polyelectrolyte layer is believed to bring very limited water transport resistance, since low-Mw polyelectrolytes were employed to avoid forming a dense layer even under a relatively high wt% of polyelectrolyte coating (*W. Jin et al.*,

J. Membr. Sci., 302, 78-86, 2007); 2) the coated polyelectrolyte can improve the hydrophilicity of membrane surface, which could offset its negative impact on transport resistance, if any, and even afford higher water permeance compared with pristine GO membrane (Fig. S22).

Fig. S22 Water permeance of surface-charged GO membranes at a function of membrane water contact angles (52.5° of GO-PEI, 48.1° of GO-PAA, 41.9° of GO-PAH, 39.1° of GO-PDDA, 33.4° of GO-PSS and 25.2° of GO-SA membranes). Solid black line: best linear fit for water permeance. Dotted blue line: water permeance of pristine GO membrane.

Comment 2

Based on the interaction energy presented in Fig 2e, how to justify the observed feed concentration dependence on the salt rejection/selectivity?

Response to Comment 2

Thanks for reviewer's question. We calculated the interaction energy in Fig. 2e to understand the effect of electrostatic interactions between charged ions and charged membrane surface that dominate the ionic transport through the surface-charged GO membrane in dilute salt solutions (*Y. Zhou et al., Carbon, 110, 56-61, 2016*). While the real process of separating ion from water is more complicated which might involve multiple other effects, such as size sieving effect and electrostatic screening effect. As increasing the salt concentration, the significantly enhanced ionic strength

would enhance the electrostatic screening effect, thereby lessening the electrostatic exclusion and decreasing salt rejection/selectivity (Fig. S12). We have revised the statement (Line 143, Page 5) in the manuscript as follow:

“In addition to the salt valence ratio that controls ion transport based on the dominant electrostatic exclusion effect in dilute salt solution, the salt concentration is another factor affecting ion transport via an electrostatic screening effect.”

Comment 3

What was the feed concentration used in Fig 3a, b?

Response to Comment 3

Thanks for the kindly reminder. The feed concentration is given in the Methods. We have added this essential information (Line 230, Page 9) in the revised manuscript as follow:

“Figure 3. Membrane performance comparison. a, Water permeance and MgCl_2 rejection of pristine GO and positively charged GO-PEI membrane as a function of membrane thickness (GO deposition amounts of 0.1, 0.2, 0.5, 0.8 and 1.0 mg with 0.1 wt% PEI surface coating) under 2 bar filtration **at feed concentration of 50 ppm**. b, Water permeance and Na_2SO_4 rejection of pristine GO and negatively charged GO-PAA membrane as a function of membrane thickness (GO deposition amounts of 0.1, 0.2, 0.5, 0.8 and 1.0 mg with 0.1 wt% PAA surface coating) under 2 bar filtration **at feed concentration of 50 ppm**.”

Comment 4

How is the surface potential of hydrated ions approximated in the model?

Response to Comment 4

Thanks for reviewer’s question. We have added the details about the calculation of surface potential of hydrated ions in the Methods (Line 456, Page 16) as below:

“The surface potential of hydrated ions is calculated according to Coulomb's law as follow:

$$\psi_s = \frac{q}{4\pi\epsilon r} \quad (18)$$

where ψ_s is surface potential of hydrated ion, q is the charge of the ion (which equals to $1e = 1.602 \times 10^{-19}C$ for monovalent ions such as Na^+ and Cl^- , and $2e = 3.204 \times 10^{-19}C$ for divalent ions such as Mg^{2+} and SO_4^{2-}), ϵ is dielectric constant, r is hydration radius of hydrated ions (which is 0.358, 0.332, 0.428, 0.379 nm for Na^+ , Cl^- , Mg^{2+} and SO_4^{2-} respectively³¹).

Ref. 31. B. Tansel, et al. Significance of hydrated radius and hydration shells on ionic permeability during nanofiltration in dead end and cross flow modes, *Sep. Purif. Technol.* **51**, 40-47 (2006).

Comment 5

Identical morphology is shown in Fig S15 for surface charged GO and layer by layer assemble sample. What was the fundamental performance enhancing factor for the surface charged GO layers over the Layer-by layer GO-polyelectrolyte membranes?

Response to Comment 5

Thanks for reviewer's question. The displayed morphologies for surface-charged GO and layer-by-layer GO membrane look similar, mainly due to the insufficient resolution under SEM to well distinguish subtle differences in nanoarchitectures of the two membrane samples, even using the highest magnification as we can. Essentially, the surface-charged GO membrane and layer-by-layer GO-polyelectrolyte membrane have intrinsically different structural characteristics determined by the distinct membrane fabrication approaches, thereby leading to disparate transport properties.

We have supplemented the XPS characterization of surface polyelectrolyte layer and underlying GO laminate in surface-charged GO membranes (Fig. S18) to investigate the chemical composition of the membranes. The characteristic peaks of N and S elements derived from the polyelectrolytes are detected only on the surface of the membranes, and are absent when etching into the underlying GO laminate. The

results clearly indicate that the polyelectrolytes only attach on top of the membrane but not enter into the GO laminates in the surface-charged GO membrane.

Moreover, we have added schematics for the three typical GO-based membranes shown in Fig. S16 to demonstrate the membrane structure more clearly. In particular, the layer-by-layer GO-polyelectrolyte membrane contains many units consisting of GO nanosheets and polyelectrolyte closely integrating with each other through strong and abundant electrostatic attraction and hydrogen bonding. Thus, a more compact laminar structure than that of pristine GO laminate would be formed, and the resulting clogged GO nanochannels would significantly increase the water transport resistance and compromise water permeance. In contrast, in the design of surface-charged GO membrane, an ultra-thin and loose polyelectrolyte layer is attached on top of GO laminate to provide surface charges only, neither introducing visible transport resistance (as explained in Response to comment 1) nor disturbing the laminar structure of underlying GO laminate (as confirmed by the XPS spectra in Fig. S18). As a result, the well-preserved GO nanochannels with intrinsically fast water transport properties enabled the surface-charged GO membrane about 4-fold higher water permeance than that of layer-by-layer GO-polyelectrolyte membrane with a similar membrane thickness and salt rejection (Fig. S17).

Fig. S16 Typical schematics and SEM cross sectional images of GO-PEI membranes with configuration of (A, D) surface-charged GO-PEI membrane (GO deposition amount of 0.5 mg with 0.1 wt% PEI solution surface coating for 30 min), (B, E) layer-by-layer GO-PEI membrane (alternately deposition of 10 mL GO suspension

containing 0.025 mg GO and 10 mL 0.1 wt% PEI solution for 20 times via vacuum filtration) and (C, F) mixed matrix GO-PEI membrane (spin coating using 20 mL mixed preparing solution containing 0.5 mg GO and 0.5 g PEI).

Fig. S17 Water permeance and MgCl₂ rejections of surface-charged, layer-by-layer, mixed matrix GO-PEI membranes.

Fig. S18 (A) XPS N1s spectrum of surface polyelectrolyte layer and underlying GO laminate in GO-PDDA membrane and (B) XPS S2p spectrum of surface polyelectrolyte layer and underlying GO laminate in GO-PSS membrane.

“Notes: As confirmed by the XPS spectra of surface polyelectrolyte layer and underlying GO laminate in surface-charged GO membranes (Fig. S18) that the characteristic peaks of N and S elements derived from the polyelectrolytes are

detected on the surface of the membranes, and are absent when etching into the underlying GO laminate. These results indicate that the polyelectrolytes only attach on top of the membrane but not enter into the GO laminates.”

Comment 6

Clarify whether the performance comparison with commercial membranes was performed at identical conditions including the feed concentration, pressure, etc.?

Response to Comment 6

Thanks for reviewer’s suggestion. The performance of commercial membranes is adopted from literature which are not exactly identical to our test conditions. To make a more accurate comparison, we have evaluated the performance of commercial membranes (using DK and DL membranes from GE as examples) under the same conditions employed in this work (feed concentration of 50 ppm and operation pressure of 2 bar). As shown in Table R1, the results fairly fall into the range summarized in Fig. 3c-d. Namely, our surface-charged GO membranes exhibit much higher water permeance and MgCl₂ rejection compared with typical commercial membranes under identical feed concentration and pressure.

We have added the feed concentration and pressure for all the membranes compared in Fig. 3c-d, and updated the additional results for commercial membranes into Table S2-3 as below:

Table R1 Water permeance with MgCl₂ and Na₂SO₄ rejections of commercial DK and DL membranes (feed concentration of 50 ppm and operation pressure of 2 bar).

Membrane	Permeance (LMH/bar)	MgCl ₂ rejection (%)	Na ₂ SO ₄ rejection (%)
DK	7.3	56.5	93.2
DL	6.2	60.7	95.8

Table S2 MgCl₂ rejection comparisons of positively charged GO membranes in this work with other nanofiltration membranes in literature.

Membrane	Method	Thickness (nm)	Feed concentration	Pressure (bar)	Permeance (LMH/bar)	Rejection (%)
Based-refluxing reduced GO/PVDF	Vacuum filtration	53	20 mM	1	3.26	20
G-CNTm(2:1)/PVDF	Vacuum filtration	40	10 mM	5	11.33	9.6
GO&EDA_HPEI 60K/PDA-PC	Pressurized filtration	69.41	1000 ppm	1	5	96
GO/PSf	Pressurized filtration	150	2000 ppm	15	11	12.5
GO/PVDF	Vacuum filtration	50	3 mM	4	2.4	19.1
TMPyP/GO/PC	Vacuum filtration	-	2000 ppm	8	11.6	45
PEI/GO/h-PAN	Layer-by-layer	59.1	1000 ppm	5	4.2	93.9
GO/PAH/h-PAN	Layer-by-layer	37	6.7 mM	6.9	2	92
Silica/polypiperazine-amide/PES	Interfacial polymerization	42	2000 ppm	6	7.8	50.7
mMSN/PA	Interfacial polymerization	100	5 mM	6	5.4	10
SMWCNT TFN	Interfacial polymerization	100	1000 ppm	6	13.2	62
PD/ZIF-8 templated PA NF	Interfacial polymerization	75	1000 ppm	4	53.5	38
Commercial NF (e.g. DK, DL)	-	-	2000 ppm	6-15	5-14	40-98
DK	-	-	50 ppm	2	7.3	56.5
DL	-	-	50 ppm	2	6.2	60.7

GO-PDDA (this work)		130			15.8	95.2
GO-PEI (this work)		130			11.4	93.1
GO-PAH (this work)	Pressurized filtration	110	50 ppm	2	13.2	77.8
GO-PDDA (long term, this work)		130			13.7	91.9
GO/TiO ₂ -PDDA (this work)		300			51.2	93.2

Table S3 Na₂SO₄ rejection comparisons of negatively charged GO membranes in this work with other nanofiltration membranes in literature.

Membrane	Method	Thickness (nm)	Feed concentration	Pressure (bar)	Permeance (LMH/bar)	Rejection (%)
TMC cross-linked GO/PSF	Layer-by-layer	14	10 mM	3.4	8-27.6	26-46
Based-refluxing reduced GO/PVDF	Vacuum filtration	53	20 mM	1	3.26	60
G-CNTm(2:1)/PVDF	Vacuum filtration	40	10 mM	5	11.33	81
GO&EDA_HPEI 60K/PDA-PC	Pressurized filtration	69.4	1000 ppm	1	5	38
GO/PSf	Pressurized filtration	150	2000 ppm	15	11	65
GO/Cellulose	Vacuum filtration	200	10 mM	-	8	67
GO/PVDF	Vacuum filtration	50	3 mM	4	2.4	79.5
GO@PAN	Vacuum filtration	128	-	1	1.8	56.7
TMPyP/GO/PC	Vacuum filtration	-	2000 ppm	8	11.6	88
PEI/GO/h-PAN	Layer-by-layer	59.1	1000 ppm	5	4.2	28
PEI/GO/h-PAN	EF assisted layer-by-layer	77	500 ppm	4	16.4	86.76
GO/ceramic membrane (TiO ₂)	Vacuum filtration	10	10 mM	3	58.6	33.7
GO/PAH/h-PAN	Layer-by-layer	37	6.7 mM	6.9	2	68

Silica/polypiperazine-amide/ PES	Interfacial polymerization	42	2000 ppm	6	7.8	97.3
mMSN/PA	Interfacial polymerization	100	5 mM	6	5.4	80
SMWCNT TFN	Interfacial polymerization	100	1000 ppm	6	13.2	96.8
MWCNT-OH TFN	Interfacial polymerization	77			6.9	97.6
MWCNT-COOH TF		84	2000 ppm	6	6.2	96.6
MWCNT-NH TFN		71			5.3	96.8
PD/ZIF-8 templated PA NF	Interfacial polymerization	75	1000 ppm	4	53.5	95
Commercial NF (e.g. DK, DL)	-	-	2000 ppm	6-15	5-14	40-98
DK	-	-	50 ppm	2	7.3	93.2
DL	-	-	50 ppm	2	6.2	95.8
GO-SA (this work)		125			20.2	88.4
GO-PAA (this work)		125			14.3	95.3
GO-PSS (this work)	Pressure-assisted filtration	120	50 ppm	2	16.8	97.1
GO-PSS (long term, this work)		120			14.7	94.6
GO/TiO ₂ -PSS (this work)		280			56.8	93.9

Comment 7

Define the solid line in Fig. 3c.

Response to Comment 7

Figs. 3c,d are summaries of performance of state-of-the-art nanofiltration membranes, and the solid lines are eye-guiding lines that are only used to indicate the general performance limit of water permeance and salt rejection in these nanofiltration membranes. We have emphasized the definition of eye-guiding lines in the caption and marked performance limit in the Figs. 3c,d (Line 242, Page 9) in the revised manuscript as follow:

Figure 3. Membrane performance comparison.

“Solid lines are eye-guiding lines only used to indicate the general performance limit of water permeance and salt rejection in nanofiltration membranes.”

In addition, we’d like to note that the comparison with a theoretical “trade-off” line between water permeability and water/salt selectivity was also given in Fig. S23 as below:

Fig. S23 Water permeability and water/salt selectivity of surface-charged GO membranes in comparison with other polymeric nanofiltration membranes. Solid line is the permeability-selectivity trade-off for polymeric nanofiltration membranes.

Comment 8

Need to justify the role of GO on the observed surface charge tuning or salt rejection properties of the studied membranes. Is the active layer GO or the

polyelectrolyte?

Response to Comment 8

Thanks for reviewer's question. The GO laminate performs three indispensable roles in the surface-charged GO membrane design. 1) GO laminate provides fast water transport channels for the membrane by utilizing the unique two-dimensional graphene capillaries (*A. K. Geim et al., Science, 2012, 335, 442-444*); 2) GO laminate with oxygen-containing groups offers plentiful interaction sites with polyelectrolyte, contributing to the firmly attachment of surface polyelectrolyte layer; 3) GO laminate offers an ideal platform with ultra-smooth surface for the uniform deposition of polyelectrolyte layer. As mentioned above, we employed low-Mw polyelectrolytes to provide surface charge only, avoiding formation of a dense layer with additional water transport resistance. This unique surface-charge design is realized by using the GO laminate as the "support". Otherwise, if directly coating on porous substrates, the low-Mw polyelectrolytes would easily fall into the substrate pores, causing serious pore penetration and thus failing to form a uniform charge layer. It is evidenced by the fact that even coating these polyelectrolytes on GO laminate with insufficient thickness, the resulting surface-charged GO membranes were unable to effectively reject salts, as shown in Figs. 3a,b. It is because the occurrence of defects in the excessively thin GO laminate (<100 nm in our case) would lead to uneven deposition of polyelectrolytes, resulting in a non-uniform distribution of surface charges and thus poor electrostatic exclusion properties of the membrane.

We have discussed the above effect (Line 217, Page 8) in the manuscript as below:

“Noting that a critical thickness of GO membrane (~100 nm in our case) is needed to provide an ideal platform with ultra-smooth and defect-free surface for the uniform deposition of polyelectrolyte layer, thereby realizing well-distributed surface charges to perform effective electrostatic exclusion function for the membrane.”

Also, we have concluded the essential roles of polyelectrolyte layer and GO laminate in Conclusion (Line 284, Page 11) in the revised manuscript as follow:

“Surface polyelectrolyte layer with tunable charge properties offered desirable

interactions with charged ions to control the ionic transport, meanwhile underlaying GO laminate with 2D graphene capillaries provided fast water transport nanochannels. By rational design of the membrane surface charge and the transport channels, the resulting surface charged GO membranes exhibited...”

Reviewer #3

General Comment

The article discusses the fabrication of graphene oxide membranes with tunable surface charge. By using different charged polyelectrolytes, the rejection in the GO membrane for divalent ions is tuned. The study is thorough, but the modification presented and the associated modeling are simple and similar methods have been performed elsewhere. Additionally, the observed impacts of surface charge on performance are predictable and well-understood. Further comments are given below.

Response to General Comment

Thanks very much for reviewer's recognition on one of the key messages of this work: the rejection of GO membrane for divalent ions is tuned by using different charged polyelectrolytes on the surface. His/her careful evaluation and detailed comments are also very helpful for us to clarify the novelty and improve the quality of our work. So far, most studies on GO membranes have focused on the manipulation of GO interlayer spacing. The novel concept of controlling surface charge of GO membrane presented in this work is distinct from the reported methods. The surface coating of selected polyelectrolytes with desirable molecular structures and charge properties on GO membrane has successfully achieved controllable ion transport without impeding water fast permeation through GO membrane. The power of the proposed surface charge controlling strategy also lies in allowing an independent structural optimization of underlying GO laminate (e.g., nanoparticles intercalation enlarging the nanochannels of GO laminate to highly enhance the water permeance demonstrated in this work), thereby realizing the breakthrough of performance limit for reported GO-based membranes and state-of-the-art nanofiltration membranes. Moreover, for the first time, we employed the model derived from classical Derjaguin-Landau-Verwey-Overbeek (DLVO) theory to reveal the ion transport mechanism through GO membrane. Prior to this work, lots of studies mainly qualitatively attributed the selective ionic transport through GO membrane to electrostatic exclusion effect. The calculation of interaction energy barrier of ions

against surface-charged membrane in this work has provided a quantitative understanding of the electrostatic exclusion effect based on electrostatic repulsion and attraction, paving a theoretical foundation for rational design of GO membrane for ion transport. More detailed justification for the novelty of this work are given in the following responses to the reviewer's comments.

Comment 1

It is well understood that surface charge in membrane will impact the rejection of charged solutes and that this effect will be reduced at higher concentrations. Additionally, several previous papers have used polyelectrolytes to create ion-selective nanofiltration membranes, including previous work with graphene oxide (<https://doi.org/10.1016/j.memsci.2014.06.036>). In this study, it is unclear how the graphene oxide base layer contributes uniquely to charge-based rejection that is observed, since the coating only impacts the surface and does not exploit the confinement between GO sheets.

Response to Comment 1

As the reviewer mentioned, several papers have reported GO-based nanofiltration membranes enabled by using polyelectrolytes (mainly via layer-by-layer assembly) for ion separation (B. Mi et al., *J. Membr. Sci.* 469, 80-87, 2014; T. Chung et al., *J. Membr. Sci.* 515, 230-237, 2016; B. Cao et al., *Appl. Surf. Sci.* 387, 521-528, 2016; S. Ji et al., *Sep. Purif. Technol.* 160, 123-131, 2016). Essentially, the surface-charged GO membrane uniquely designed in this work has intrinsically different structural characteristics from the reported layer-by-layer GO-polyelectrolyte membrane owing to the distinct membrane fabrication approaches, thereby leading to disparate transport properties.

We have supplemented the XPS characterization of surface polyelectrolyte layer and underlying GO laminate in surface-charged GO membranes (Fig. S18) to investigate the chemical composition of the membranes. The characteristic peaks of N and S elements derived from the polyelectrolytes are detected on the surface of the membranes, and are absent when etching into the underlying GO laminate. The

results clearly indicate that the polyelectrolytes only attach on top of the membrane but not enter into the GO laminates in the surface-charged GO membrane.

We have added schematics for the three typical GO-based membranes shown in Fig. S16 to demonstrate the membrane structure more clearly. In particular, the layer-by-layer GO-polyelectrolyte membrane contains many units consisting of GO nanosheets and polyelectrolyte closely integrating with each other through strong and abundant electrostatic attraction and hydrogen bonding. Thus, a more compact laminar structure than that of pristine GO laminate will be formed, and the resulting clogged GO nanochannels would significantly increase water transport resistance and compromise water permeance. In contrast, in the design of surface-charged GO membrane, an ultra-thin and loose polyelectrolyte layer is attached on top of GO laminate to provide surface charges only, without introducing visible transport resistance (as shown in Fig. 3a,b) and disturbing the laminar structure of underlying GO laminate (as confirmed by the XPS spectra in Fig. S18). As a result, the well-preserved GO nanochannels with intrinsically fast water transport properties enabled the surface-charged GO membrane about 4-fold higher water permeance than that of layer-by-layer GO-polyelectrolyte membrane with a similar membrane thickness and salt rejection (Fig. S17). Namely, the novel design of surface-charged GO membrane has successfully achieved outstanding salt rejection and meanwhile ultrahigh water permeance for nanofiltration process which is rarely realized for the reported GO membranes.

Fig. S16 Typical schematics and SEM cross sectional images of GO-PEI membranes

with configuration of (A, D) surface-charged GO-PEI membrane (GO deposition amount of 0.5 mg with 0.1 wt% PEI solution surface coating for 30 min), (B, E) layer-by-layer GO-PEI membrane (alternately deposition of 10 mL GO suspension containing 0.025 mg GO and 10 mL 0.1 wt% PEI solution for 20 times via vacuum filtration) and (C, F) mixed matrix GO-PEI membrane (spin coating using 20 mL mixed preparing solution containing 0.5 mg GO and 0.5 g PEI).

Fig. S17 Water permeance and MgCl₂ rejections of surface-charged, layer-by-layer, mixed matrix GO-PEI membranes.

Fig. S18 (A) XPS N1s spectrum of surface polyelectrolyte layer and underlying GO laminate in GO-PDDA membrane and (B) XPS S2p spectrum of surface polyelectrolyte layer and underlying GO laminate in GO-PSS membrane.

“Notes: As confirmed by the XPS spectra of surface polyelectrolyte layer and underlying GO laminate in surface-charged GO membranes (Fig. S18) that the characteristic peaks of N and S elements derived from the polyelectrolytes are detected on the surface of the membranes, and are absent when etching into the underlying GO laminate. These results indicate that the polyelectrolytes only attach on top of the membrane but not enter into the GO laminates.”

The GO base layer performs three indispensable roles in the surface-charged GO membrane design. 1) GO laminate provides fast water transport channels for the membrane by utilizing the unique two-dimensional graphene capillaries (*A. K. Geim et al., Science, 2012, 335, 442-444*); 2) GO laminate with oxygen-containing groups offers plentiful interaction sites with polyelectrolyte, contributing to the firmly attachment of surface polyelectrolyte layer; 3) GO laminate offers an ideal platform with ultra-smooth surface for the uniform deposition of polyelectrolyte layer. We employed low-Mw polyelectrolytes to provide surface charge only, avoiding formation of a dense layer with additional water transport resistance (*W. Jin et al., J. Membr. Sci., 302, 78-86, 2007*). This unique surface-charge design is realized by using the GO laminate as the “support”. Otherwise, if directly coating on porous substrates, the low-Mw polyelectrolytes would easily fall into the substrate pores, causing serious pore penetration and thus failing to form a uniform charge layer. It is evidenced by the fact that even coating these polyelectrolytes on GO laminate with insufficient thickness, the resulting surface-charged GO membranes were unable to effectively reject salts, as shown in Figs. 3a,b. It is because the occurrence of defects in the excessively thin GO membrane (<100 nm in our case) would lead to uneven deposition of polyelectrolytes, resulting in a non-uniform distribution of surface charges and thus poor electrostatic exclusion properties of the membrane.

We have discussed the above effect (Line 217, Page 8) in the manuscript as below:

“Noting that a critical thickness of GO membrane (~100 nm in our case) is needed to provide an ideal platform with ultra-smooth and defect-free surface for the

uniform deposition of polyelectrolyte layer, thereby realizing well-distributed surface charges to perform effective electrostatic exclusion function for the membrane.”

Also, we have concluded the essential roles of polyelectrolyte layer and GO laminate in Conclusion (Line 284, Page 11) in the revised manuscript as follow:

“Surface polyelectrolyte layer with tunable charge properties offered desirable interactions with charged ions to control the ionic transport, meanwhile underlaying GO laminate with 2D graphene capillaries provided fast water transport nanochannels. By rational design of the membrane surface charge and the transport channels, the resulting surface charged GO membranes exhibited...”

Comment 2

Figure 3: The trade-off curves drawn here appear to be arbitrary. The conventional trade-off for polymeric membranes is drawn with permeability of the solvent and solute on different axes; this method is useful because permeability is close to an intrinsic material property. As it is currently drawn in Fig 3, the curve does not appear to have much fundamental basis. Why should the trade-off be a linear curve with rejection on the y-axis? Also, comparing solely based on rejection is not justifiable since the effect of salt concentration and operating conditions is not accounted for.

Response to Comment 2

Thanks for reviewer’s comments. We fully agree with the reviewer that polymeric membranes usually suffer the conventional permeability-selectivity trade-off relationship. We discussed this theoretical trade-off in Fig. S23 shown as below. The water permeance and salt rejection are respectively normalized to water permeability and water/salt selectivity according to the solution-diffusion model to eliminate the effects of membrane thickness and operating conditions such as feed pressure and salt concentration (*H. Park et al., J. Membr. Sci. 369, 130-138, 2011*). As the reviewer pointed out, such trade-off analysis provides a clear comparison of intrinsic properties of the membrane materials, which has been widely used in literature (*B. D. Freeman et al., Science, 356, eaab0530, 2017; H. Park et al., J. Membr. Sci. 544, 425-435,*

2017; L. Zhang *et al.*, *Science*, 360, 518-521, 2018). As demonstrated in Fig. S23, the surface charged GO membranes proposed in this work exhibit both high water permeability and high water/salt selectivity, fairly transcending the upper-bound for traditional polymeric membranes.

Fig. S23 Water permeability and water/salt selectivity of surface-charged GO membranes in comparison with other polymeric nanofiltration membranes. Solid line is the permeability-selectivity trade-off for polymeric nanofiltration membranes.

Figs. 3c,d are summaries of the separation performance (i.e., water permeance and salt rejection rate) of state-of-the-art nanofiltration membranes for typical salts rejection. Such performance comparison is frequently used to reflect the actual separation performance of membranes rather than intrinsic transport property of membrane materials (*J. C. Grossman et al.*, *Nano letters*, 12, 3602-3608, 2012; *H. Jung et al.*, *J. Mater. Chem. A*, 4, 17773-17781, 2016; *H. Wang et al.*, *Angew. Chem. Int. Ed.*, 56, 1825-1829, 2017; *Q. Zhang et al.*, *J. Mater. Chem. A*, 5, 14819-14827, 2017; *J. Jin et al.*, *Nat. Comm.* 9, 2004, 2018). The solid lines in Figs. 3c,d are eye-guiding lines only used to indicate the general performance limit of water permeance and salt rejection in nanofiltration membranes. We have emphasized the definition of eye-guiding lines in the caption and marked performance limit in the Figs. 3c,d in the revised manuscript as follow:

Figure 3. Membrane performance comparison.

“Solid lines are eye guidelines used only to indicate the general performance limit of permeance and rejection in nanofiltration membranes.”

As pointed out by the reviewer, the effects of salt concentration and operating conditions are important factors for determining the membrane separation performance. Therefore, we have added the feed concentration and pressure for all the membranes compared in Fig. 3c-d into Table S2-3 as below. Moreover, we have evaluated the performance of available membranes (using commercial DK and DL membranes from GE as examples) under the same conditions in this work. As expected, the results fall into the range summarized in Fig. 3c-d. Namely, our surface-charged GO membranes exhibit much higher water permeance and MgCl₂ rejection compared with typical commercial membranes under identical feed concentration and pressure.

Table S2 MgCl₂ rejection comparisons of positively charged GO membranes in this work with other nanofiltration membranes in literature.

Membrane	Method	Thickness (nm)	Feed concentration (mM)	Pressure (bar)	Permeance (LMH/bar)	Rejection (%)
Based-refluxing reduced GO/PVDF	Vacuum filtration	53	20 mM	1	3.26	20
G-CNTm(2:1)/PVDF	Vacuum filtration	40	10 mM	5	11.33	9.6

GO&EDA_HPEI 60K/PDA-PC	Pressurized filtration	69.41	1000 ppm	1	5	96
GO/PSf	Pressurized filtration	150	2000 ppm	15	11	12.5
GO/PVDF	Vacuum filtration	50	3 mM	4	2.4	19.1
TMPyP/GO/PC	Vacuum filtration	-	2000 ppm	8	11.6	45
PEI/GO/h-PAN	Layer-by-layer	59.1	1000 ppm	5	4.2	93.9
GO/PAH/h-PAN	Layer-by-layer	37	6.7 mM	6.9	2	92
Silica/polypiperazine-amide/ PES	Interfacial polymerization	42	2000 ppm	6	7.8	50.7
mMSN/PA	Interfacial polymerization	100	5 mM	6	5.4	10
SMWCNT TFN	Interfacial polymerization	100	1000 ppm	6	13.2	62
PD/ZIF-8 templated PA NF	Interfacial polymerization	75	1000 ppm	4	53.5	38
Commercial NF (e.g. DK, DL)	-	-	2000 ppm	6-15	5-14	40-98
DK	-	-	50 ppm	2	7.3	56.5
DL	-	-	50 ppm	2	6.2	60.7
GO-PDDA (this work)		130			15.8	95.2
GO-PEI (this work)		130			11.4	93.1
GO-PAH (this work)	Pressurized filtration	110	50 ppm	2	13.2	77.8
GO-PDDA (long term, this work)		130			13.7	91.9
GO/TiO ₂ -PDDA (this work)		300			51.2	93.2

Table S3 Na₂SO₄ rejection comparisons of negatively charged GO membranes in this work with other nanofiltration membranes in literature.

Membrane	Method	Thickness	Feed	Pressur	Permeanc	Rejection
----------	--------	-----------	------	---------	----------	-----------

		Thickness (nm)	Concentration	Pressure (bar)	Flow rate (LMH/bar)	Retention (%)
TMC cross-linked GO/PSF	Layer-by-layer	14	10 mM	3.4	8-27.6	26-46
Based-refluxing reduced GO/PVDF	Vacuum filtration	53	20 mM	1	3.26	60
G-CNTm(2:1)/PVDF	Vacuum filtration	40	10 mM	5	11.33	81
GO&EDA_HPEI 60K/PDA-PC	Pressurized filtration	69.4	1000 ppm	1	5	38
GO/PSf	Pressurized filtration	150	2000 ppm	15	11	65
GO/Cellulose	Vacuum filtration	200	10 mM	-	8	67
GO/PVDF	Vacuum filtration	50	3 mM	4	2.4	79.5
GO@PAN	Vacuum filtration	128	-	1	1.8	56.7
TMPyP/GO/PC	Vacuum filtration	-	2000 ppm	8	11.6	88
PEI/GO/h-PAN	Layer-by-layer	59.1	1000 ppm	5	4.2	28
PEI/GO/h-PAN	EF assisted layer-by-layer	77	500 ppm	4	16.4	86.76
GO/ceramic membrane (TiO ₂)	Vacuum filtration	10	10 mM	3	58.6	33.7
GO/PAH/h-PAN	Layer-by-layer	37	6.7 mM	6.9	2	68
Silica/polypiperazine-amide/PES	Interfacial polymerization	42	2000 ppm	6	7.8	97.3
mMSN/PA	Interfacial polymerization	100	5 mM	6	5.4	80
SMWCNT TFN	Interfacial polymerization	100	1000 ppm	6	13.2	96.8
MWCNT-OH TFN	Interfacial polymerization	77	2000 ppm	6	6.9	97.6
MWCNT-COOH TFN	Interfacial polymerization	84			6.2	96.6

MWCNT-NH TFN	n	71			5.3	96.8
PD/ZIF-8 templated PA NF	Interfacial polymerization	75	1000 ppm	4	53.5	95
Commercial NF (e.g. DK, DL)	-	-	2000 ppm	6-15	5-14	40-98
DK	-	-	50 ppm	2	7.3	93.2
DL	-	-	50 ppm	2	6.2	95.8
GO-SA (this work)		125			20.2	88.4
GO-PAA (this work)		125			14.3	95.3
GO-PSS (this work)	Pressure-assisted filtration	120	50 ppm	2	16.8	97.1
GO-PSS (long term, this work)		120			14.7	94.6
GO/TiO ₂ -PSS (this work)		280			56.8	93.9

Comment 3

Line 33: The frictionless flow in graphene oxide membranes has been disputed by a number of studies (DOI: 10.1021/acsnano.8b02015).

Response to Comment 3

Thanks for reviewer's comments. Indeed, there is no clear agreement on the water transport pathways and mechanisms through GO laminates. Thus, we have revised the statement (Line 33, Page 2) in the manuscript as follow:

“Graphene oxide (GO) membrane is expected to share structural features with biological membrane owing to its fast water-transport pathways through assembled GO laminates, which has generated immense interest from the scientific community to study its exceptional transport properties⁷⁻⁹.”

Ref. 9. V. Saraswat et al. Invariance of water permeance through size-differentiated graphene oxide laminates, *ACS Nano* **12**, 7855-7865 (2018).

Reviewer #4

General Comment

The rejection of GO membranes for ions under pressure remains a big challenge. In this work, the authors achieved an exceptional enhancement of ion rejection without compromising GO membranes' water permeance by manipulating the membrane surface charge properties. The idea is great and experiments carried out are comprehensive. Still, a few questions.

Response to General Comment

We are grateful for the positive comment on our work. All the revised parts can be found in the revised manuscript with changes marked.

Comment 1

In page 3, line 67-70, the photograph shows a large area GO membrane (diameter 15 cm) with GO loading amount of 5 mg. The SEM image is for a GO membrane with GO loading amount of 0.5 mg. I guess the diameter should be 4.7 cm. Also, the captions for Fig. S2 should be modified to avoid confusion.

Response to Comment 1

Thanks for the kindly reminder. The diameters of surface charged GO membranes in Fig. 1d and S2 are 4.7 cm with effective membrane area of $\sim 17.35 \text{ cm}^2$. We have revised the captions for Fig. 1 (Line 75, Page 3) and Fig. S2 as follow:

“d, SEM cross-sectional views of surface-charged GO membranes on top of a porous PAN substrate (GO deposition amount of 0.5 mg with 0.1 wt% PEI polyelectrolyte surface coating; **membrane diameter of 4.7 cm with effective area of $\sim 17.35 \text{ cm}^2$**).”

“SEM surface morphology and cross-sectional view of (A) pristine GO membrane (GO deposition amount of 0.5 mg; Insert: SEM image of PAN substrate surface; **membrane diameter of 4.7 cm with effective area of $\sim 17.35 \text{ cm}^2$**) and (B-G) surface-charged GO membranes (GO deposition amount of 5 mg with 0.1 wt% PDDA, PEI, PAH, SA, PAA, PSS polyelectrolytes surface coating respectively; **membrane**

diameter of 4.7 cm with effective area of $\sim 17.35 \text{ cm}^2$).”

Comment 2

Is the solution-diffusion model suitable for GO membranes? It is widely used for RO membranes. For NF membranes where electrostatic interaction is an important consideration, the Donnan equilibrium model and the extended Nernst-Planck model may work better. What's more, GO membranes differ a lot from polymeric NF membranes. The nanocapillaries are horizontal, not vertical. These models also work in this case?

Response to Comment 2

Thanks for reviewer's comments. The transport model for GO membranes remains under investigation by several groups. The solution-diffusion model has been widely used to understand transport mechanism through membranes with pore size of around or less than 1 nm used for separating water from saline solution or gas mixtures (R. W. Baker et al., *J. Membr. Sci.* 107, 1-21, 1995). As the reviewer pointed out, the electrostatic interaction is indeed an important consideration for nanofiltration membranes. In this work, we consider such electrostatic attraction and/or repulsion as a second-order effect (ion sorption/solution) relative to the solution-diffusion model. Compared with the Donnan equilibrium model and the extended Nernst-Planck model, which mainly describe the charge-guided ion transport, the solution-diffusion model could provide a more comprehensive description on the chemical potential, concentration, and pressure gradient-driven ion transport and water permeation behaviors in the membrane (B. D. Freeman et al., *Prog. Polym. Sci.* 39, 1-42, 2014). In addition, the solution-diffusion model works well for polymeric nanofiltration membranes, whose nanocapillaries (e.g., free-volume cavities) for molecular transport are generally disordered, including vertical and horizontal channels. It is reasonable to believe that the solution-diffusion model can be applied for GO membranes consisting of horizontal nanocapillaries and vertical slit-like pores.

Comment 3

The authors found that the NaCl transport behaviors were almost unchanged with positively and negatively charged GO membranes because Z^+/Z is 1 for NaCl. Can any explanations be provided for the difference in the permeability of $MgSO_4$?

Response to Comment 3

Thanks for reviewer's question. Although there is a balanced electrostatic interaction with co-ions and counter-ions in the case of $MgSO_4$ with $Z^+/Z=1$, the difference in hydrated ionic radius of Mg^{2+} (0.428 nm) and SO_4^{2-} (0.397 nm) would also affect the permeation of $MgSO_4$ through the surface-charged GO membranes. The positively charged GO membrane tends to repel Mg^{2+} and attract SO_4^{2-} , in which the smaller SO_4^{2-} is easier to pass through the membrane thus leading to a relatively higher salt permeability. Conversely, the larger Mg^{2+} yields greater steric hindrance to transport through the negatively charged GO membrane, resulting in a relatively lower salt permeability. We have added the above discussions (Line 140, Page 5) in the manuscript and Supplementary note for Fig. S11 as below:

“The variation of $MgSO_4$ permeability in positively and negatively charged GO membranes reflects the additional size discrimination effect on the ionic transport (detailed discussion can be found in Supplementary note for Fig. S11).”

Notes for Fig. S11: ...Although there is a balanced electrostatic interaction with co-ions and counter-ions in the case of $MgSO_4$ with $Z^+/Z=1$, the difference in hydrated ionic radius of Mg^{2+} (0.428 nm) and SO_4^{2-} (0.397 nm) would also affect the permeation of $MgSO_4$ through the surface-charged GO membranes. The positively charged GO membrane tends to repel Mg^{2+} and attract SO_4^{2-} , in which the smaller SO_4^{2-} is easier to pass through the membrane thus leading to a relatively higher salt permeability. Conversely, the larger Mg^{2+} yields greater steric hindrance to transport through the negatively charged GO membrane, resulting in a relatively lower salt permeability.

Comment 4

It can be concluded from the authors' findings that the surface charge properties

of GO membranes dominate their ion rejection performance. Fig. 3(a-b) shows that GO membranes' rejections for MgCl₂ and Na₂SO₄ are low when their thicknesses are smaller than 125 nm, even if they are charged. Can any possible reasons be provided?

Response to Comment 4

As the reviewer observed, GO membranes with thickness thinner than ~125 nm show relatively low salt rejections, which can be ascribed to the occurrence of defects in these excessively thin GO membranes. These defects in GO membrane would result in an uneven attachment of surface polyelectrolytes, thereby leading to a non-uniform distribution of surface charges and thus the poor salt rejection of the membrane. Therefore, GO membrane needs to reach a critical thickness to provide an ideal platform for uniform surface coating of polyelectrolyte. As suggested, we have discussed the above effect (Line 217, Page 8) in the manuscript as below:

“Noting that a critical thickness of GO membrane (~100 nm in our case) is needed to provide an ideal platform with ultra-smooth and defect-free surface for the uniform deposition of polyelectrolyte layer, thereby realizing well-distributed surface charges to perform effective electrostatic exclusion function for the membrane.”

Comment 5

I'm interested in the long period testing of the membranes. Assuming that the testing was carried out under 1 bar, water flux of the membranes as 15 LMH/bar, effective filtration area as 17.35 cm², the volume of the filtrate should be more than 3 L. A huge feed container was used? Or, the testing was carried out using a cross-flow system?

Response to Comment 5

Thanks for reviewer's question. The long-term operation for performance measurement of surface charged GO membranes employs semi-continuous process. As schematically in Fig. R3a, we recycled the permeation into the feed tank during operation to maintain a stable salt concentration in the feed. Otherwise, the feed salt concentration would continuously increase and cause concentration polarization. As demonstrated in Fig. S12, the salt concentration could affect the membrane

performance. Thus, we controlled the time interval of permeation recycling within 3 hours to keep a limited salt concentration variation of 50-72 ppm in the feed (Fig. R3b) that is proven to show little influence on the separation performance according to Fig. S12. We have added the detailed membrane preparation description (Line 368, Page 13) in the Method as follow:

“During the filtration process, we recycled the permeation into the feed tank to maintain a stable salt concentration in the feed.”

Fig. R3 (A) Schematic of long-term operation for semi-continuous performance measurement and (B) Variation of feed concentration ranging from 50 ppm to 72 ppm with operation time. Feed volume is 500 mL, operation pressure is 2 bar, and effective filtration area is 17.35 cm².

REVIEWERS' COMMENTS:

Reviewer #1 (Remarks to the Author):

Items raised have been deeply satisfied, the manuscript can be now accepted.

Reviewer #2 (Remarks to the Author):

The authors addressed all my questions and I support publication.

Reviewer #3 (Remarks to the Author):

The authors have responded well to many of the comments provided by the reviewers. The manuscript was already thorough, and the additional changes make it very rigorous. However, some of my comments (which were also discussed by other reviewers) have still not been sufficiently addressed.

The lines drawn in Fig. 3 are problematic. In the permeability and selectivity curves used by Robeson and Freeman (among others), the slope of the curve is held constant and tied to fundamental parameters with a mathematical relationship described in the literature (DOI: 10.1016/j.memsci.2010.11.054, DOI: 10.1021/es203197e). Given this relationship, one would assume the lines drawn in Fig. 3 would be nonlinear and bear similar shapes. The curves drawn in Fig. 3 are arbitrarily linear and do not have a consistent slope or real basis in the existing data. The authors must either remove the lines or carry out a proper analysis to draw these curves.

It should be noted that a water permeability coefficient exceeding 5 LMH/bar is not desirable for brackish water desalination, and even lower water permeabilities (2-3 LMH/bar) are appropriate for seawater reverse osmosis (DOI: 10.1021/acs.estlett.6b00050).

Figure S17: Rejection plotted with a scale from 0-100% makes it very difficult to understand differences in performance between these membranes. If possible, salt permeability should be used instead.

The claim of ultrafast transport in 2D graphene capillaries that the authors make in the manuscript and rebuttal (based on the Geim 2012 paper) has been contested by many studies (DOI: 10.1103/PhysRevE.89.012113 DOI: 10.1038/srep29484).

Reviewer #4 (Remarks to the Author):

The authors have addressed the questions and concerns and the paper can be accepted.

Responses to Reviewer' Comments

Manuscript #: NCOMMS-18-33220A

Title: Controllable ion transport by surface-charged graphene oxide membrane

Reviewer #3

General Comment

The authors have responded well to many of the comments provided by the reviewers. The manuscript was already thorough, and the additional changes make it very rigorous. However, some of my comments (which were also discussed by other reviewers) have still not been sufficiently addressed.

Response to General Comment

We are grateful for the reviewer's positive comments.

Comment 1

The lines drawn in Fig. 3 are problematic. In the permeability and selectivity curves used by Robeson and Freeman (among others), the slope of the curve is held constant and tied to fundamental parameters with a mathematical relationship described in the literature (DOI: 10.1016/j.memsci.2010.11.054, DOI: 10.1021/es203197e). Given this relationship, one would assume the lines drawn in Fig. 3 would be nonlinear and bear similar shapes. The curves drawn in Fig. 3 are arbitrarily linear and do not have a consistent slope or real basis in the existing data. The authors must either remove the lines or carry out a proper analysis to draw these curves.

Response to Comment 1

Thanks for reviewer's suggestion. We have removed the eye-guiding lines in Fig.3c and d, and revised the legend accordingly in the manuscript as follow:

Fig. 3 Membrane performance comparison. **a**, Water permeance and MgCl₂ rejection of pristine GO and positively charged GO-PEI membrane as a function of membrane thickness (GO deposition amounts of 0.1, 0.2, 0.5, 0.8 and 1.0 mg with 0.1 wt% PEI surface coating) under 2 bar filtration at feed concentration of 50 ppm. **b**, Water permeance and Na₂SO₄ rejection of pristine GO and negatively charged GO-PAA membrane as a function of membrane thickness (GO deposition amounts of 0.1, 0.2, 0.5, 0.8 and 1.0 mg with 0.1 wt% PAA surface coating) under 2 bar filtration at feed concentration of 50 ppm. Dashed lines: pristine GO membranes; solid lines: surface-charged GO membranes. Yellow and green upward arrows indicate the remarkable improvements of surface-charged GO membranes in salt rejection. Error bars represent standard deviations for 3 measurements. **c**, MgCl₂ rejection with water permeance of positively charged GO membranes (GO-PDDA, GO-PEI, GO-PAH, TiO₂ intercalated GO-PDDA). **d**, Na₂SO₄ rejection with water permeance of negatively charged GO membranes (GO-SA, GO-PAA, GO-PSS, TiO₂ intercalated GO-PSS) in this work, as well as comparison with 2D-material membranes (marked as squares), thin film nanocomposite (TFN) and/or thin film composite (TFC) membranes (marked as circles), and commercial polymeric nanofiltration membranes (marked as gray regions, e.g., NF270, NF90, NF200 membranes from Dow; DK, DL series of membranes from GE; ESNA series of membranes from Hydranautics). References see Supplementary Table 2 and 3 in detail.

Comment 2

It should be noted that a water permeability coefficient exceeding 5 LMH/bar is not desirable for brackish water desalination, and even lower water permeabilities (2-3 LMH/bar) are appropriate for seawater reverse osmosis (DOI: 10.1021/acs.estlett.6b00050).

Response to Comment 2

Thanks for reviewer's comment. In this work, we manipulated surface charge of GO membranes to

achieve high salts rejection while maintaining high water permeance, mainly focusing on studying the selective ions transport behaviors and mechanisms as function of membrane surface charge properties. We agree with the reviewer that when the salts rejection is very high (>99.5%), increased water permeance may have limited impact on the efficiency of desalination processes. We have added some discussions and cited relevant literature in the revised manuscript to address this issue as follow:

“In real water treatment applications, appropriate permeability (water permeance) as well as enhanced selectivity (salt rejection) are critically required for high efficiency of desalination processes²⁸.”
 Ref. 28 J. R. Werber et al. The critical need for increased selectivity, not increased water permeability, for desalination membranes, *Environ. Sci. Technol. Lett.* **3**, 112-120 (2016).

Comment 3

Figure S17: Rejection plotted with a scale from 0-100% makes it very difficult to understand differences in performance between these membranes. If possible, salt permeability should be used instead.

Response to Comment 3

Thanks for reviewer’s suggestion. We have calculated water permeability and salt permeability of the three membranes and added the results in Supplementary Figure 17 as follow:

Supplementary Figure 17 (A) Water permeance and MgCl₂ rejections, and **(B)** water permeability and salt permeability of surface-charged, layer-by-layer, mixed matrix GO-PEI membranes.

“The MgCl₂ rejections of surface-charged, layer-by-layer, mixed matrix GO-PEI membranes keep as high as ~95% and the salt permeabilities of them are lower than $3 \times 10^{-11} \text{ cm}^2 \text{ s}^{-1}$, because the introduction of PEI providing positively charged membranes surface. However, the water permeance of surface-charged GO-PEI membrane is significantly higher than two other membranes with almost double

the water permeability of that of layer-by-layer, mixed matrix GO-PEI membranes, which indicates the superiority of our surface charge controlling strategy. The low water permeance of other membrane configurations is due to no fully utilizing of channel-facilitated water transport within GO laminate.”

Comment 4

The claim of ultrafast transport in 2D graphene capillaries that the authors make in the manuscript and rebuttal (based on the Geim 2012 paper) has been contested by many studies (DOI: 10.1103/PhysRevE.89.012113 DOI: 10.1038/srep29484).

Response to Comment 4

Thanks for reviewer’s comment. We agree with the reviewer that there are disputes on ultrafast transport in 2D graphene capillaries, and there is still no clear agreement on the water transport pathways and mechanisms through GO laminates. We have cited relevant literatures and revised our statement in the manuscript as follow:

“Graphene oxide (GO) membrane is expected to share structural features with biological membrane owing to its water-transport pathways through assembled GO laminates, which has generated immense interest from the scientific community to study its transport properties and mechanisms⁷⁻¹¹.”

Ref. 10 N. Wei, X. Peng, Z. Xu. Breakdown of fast water transport in graphene oxides, *Phys. Rev. E* **89**, 012113 (2014).

Ref. 11 R. Devanathan, et al. Molecular dynamics simulations reveal that water diffusion between graphene oxide layers is slow, *Sci. Rep.* **6**, 29484 (2016).